# Coordination of planar cell polarity pathways through Spiny-legs

**Abhijit A Ambegaonkar[1,2,3], Kenneth D Irvine[1,2,3]***

[1]Howard Hughes Medical Institute, Rutgers University, Piscataway, United States;
[2]Waksman Institute of Microbiology, Rutgers University, Piscataway, United States;
[3]Department of Molecular Biology and Biochemistry, Rutgers University, Piscataway, United States

**Abstract** Morphogenesis and physiology of tissues and organs requires planar cell polarity (PCP) systems that orient and coordinate cells and their behaviors, but the relationship between PCP systems has been controversial. We have characterized how the Frizzled and Dachsous-Fat PCP systems are connected through the Spiny-legs isoform of the Prickle-Spiny-legs locus. Two different components of the Dachsous-Fat system, Dachsous and Dachs, can each independently interact with Spiny-legs and direct its localization in vivo. Through characterization of the contributions of Prickle, Spiny-legs, Dachsous, Fat, and Dachs to PCP in the *Drosophila* wing, eye, and abdomen, we define where Dachs-Spiny-legs and Dachsous-Spiny-legs interactions contribute to PCP, and provide a new understanding of the orientation of polarity and the basis of PCP phenotypes. Our results support the direct linkage of PCP systems through Sple in specific locales, while emphasizing that cells can be subject to and must ultimately resolve distinct, competing PCP signals.

***For correspondence:** irvine@ waksman.rutgers.edu

**Competing interests:** The authors declare that no competing interests exist

## Introduction

Planar cell polarity (PCP) is the coordinated orientation of cell structures and behaviors within the plane of a tissue. Manifestations of PCP include the orientation of hairs, bristles, stereocilia, and ommatidia, as well as orientated cell divisions and cell movements. Two conserved molecular systems play key roles in the establishment and maintenance of PCP: the Frizzled (Fz) PCP pathway and the Dachsous (Ds)-Fat PCP pathway (*Goodrich and Strutt, 2011*; *Matis and Axelrod, 2013*). These involve distinct components, but share common features, including the polarized localization of key components within cells, and the existence of asymmetric intercellular interactions that enable this polarity to be propagated from cell to cell. Many processes are influenced by both PCP systems, but the relationship between them has been controversial.

The *Drosophila* Ds-Fat PCP pathway includes the cadherin family proteins Ds and Fat, which interact between neighboring cells (*Ma et al., 2003*; *Matakatsu and Blair, 2004*; *Strutt and Strutt, 2002*). Binding between Ds and Fat is modulated by Four-jointed (Fj), a Golgi-localized kinase that phosphorylates their cadherin domains (*Brittle et al., 2010*; *Ishikawa et al., 2008*; *Simon et al., 2010*). Ds and Fj are expressed in opposing gradients, which orient Ds-Fat PCP (*Casal et al., 2002*; *Mao et al., 2006*; *Strutt and Strutt, 2002*; *Yang et al., 2002*; *Zeidler et al., 1999*; *2000*). Fat protein in a cell within a Ds-Fj gradient preferentially accumulates along the side where it contacts cells with higher Ds and lower Fj; Ds protein localizes in a complementary orientation (*Figure 1A*) (*Ambegaonkar et al., 2012*; *Bosveld et al., 2012*; *Brittle et al., 2012*). How polarization of Ds and Fat proteins establishes PCP is incompletely understood, but it is achieved in part through the unconventional myosin Dachs, whose membrane localization is regulated by Fat (*Mao et al., 2006*). Mammalian homologues of Ds and Fat, Dchs1 and Fat4, also influence PCP (*Mao et al., 2011a*;

**eLife digest** Animals have many asymmetric organs. Wings, for example, are aerodynamically shaped and have a clear front, back, top and bottom, and even additions to these organs, such as feathers on the wing, often need to be oriented in a specific manner.

This kind of orientation arises when cells divide and grow asymmetrically in a flat plane. The asymmetry is established at the level of single cells when proteins are not equally spread throughout a cell, but rather asymmetrically distributed. Such cells are said to be 'planar polarized'; and many experiments addressing this so-called planar cell polarity have been conducted in fruit flies, because they can be genetically altered easily. Previous studies have shown that two signaling pathways—called Frizzled and Dachsous-Fat—regulate how individual cells orient themselves within a flat sheet of cells that forms fruit fly's wing. The two pathways are not independent, but it is unclear how they are linked. In particular, there has been conflicting evidence as to whether the Dachsous-Fat pathway controls the Frizzled pathway or whether the two act in parallel.

Now, Ambegaonkar and Irvine have discovered new roles for a protein that is involved in both pathways, called 'Spiny-legs'. This protein was known to be important in the Frizzled pathway, but, when it was tracked with a fluorescent tag in developing wing cells it also accumulated in areas where two proteins that make up part of the Dachsous-Fat pathway were located. Biochemical experiments showed that both of these proteins (which are called Dachs or Dachsous) could physically interact with Spiny-legs. Ambegaonkar and Irvine therefore deleted the genes for Dachs or Dachsous in fruit flies and observed that Spiny-legs no longer organized itself in the proper way, implying that Dachs and Dachsous control where Spiny-legs goes within cells.

When this analysis was extended to other fruit fly organs, such as the eyes, Ambegaonkar and Irvine found that Dachsous was more important than Dachs for the correct localization of Spiny-legs. Additionally, the Frizzled and Dachsous-Fat pathways seemed to compete for interactions with Spiny-legs. This connection between the two pathways helps to explain how cells behave when several different signals reach them. It also shows how different organs can reuse conserved components of the pathways to make different end products. Future studies should aim to work out the number of systems that polarize cells and how they are connected in different tissues.

*Saburi et al., 2008*; *Zakaria et al., 2014*), and human FAT4 can rescue PCP phenotypes of *Drosophila fat* mutants (*Pan et al., 2013*).

The *Drosophila* Fz PCP pathway includes the asymmetrically distributed transmembrane proteins Fz and Van Gogh (Vang, also known as Strabismus), which act together with the cadherin family protein Starry night (Stan, also known as Flamingo) (*Figure 1A*) (*Chae et al., 1999*; *Goodrich and Strutt, 2011*; *Park et al., 1994*; *Shimada et al., 2001*; *Strutt, 2001*; *Usui et al., 1999*; *Vinson and Adler, 1987*). Stan interacts with Stan-Fz heterodimers in neighboring cells (*Chen et al., 2008*; *Struhl et al., 2012*; *Strutt and Strutt, 2008*); interactions between Vang and Fz have also been reported (*Wu and Mlodzik, 2008*). Each of these transmembrane complexes is associated with distinct cytoplasmic proteins, Fz-Stan associates with Dishevelled (Dsh) and Diego (Dgo) (*Axelrod, 2001*; *Feiguin et al., 2001*; *Shimada et al., 2001*; *Strutt, 2001*). Vang-Stan associates with Prickle-Spiny legs (Pk-Sple) (*Bastock et al., 2003*; *Jenny et al., 2003*). Polarization of these protein complexes can propagate from cell to cell through heterophilic intercellular interaction between Stan and Fz-Stan complexes (*Chen et al., 2008*; *Goodrich and Strutt, 2011*; *Struhl et al., 2012*). Polarization of complexes within a cell is reinforced by inhibitory intracellular interactions between asymmetrically localized components (*Goodrich and Strutt, 2011*). Mutation of any of the core members of the Fz PCP system will impair the polarization of all of the others, emphasizing their mutual dependency for polarization (*Strutt and Strutt, 2009*).

Genetic interactions between Ds-Fat pathway genes and Fz pathway genes, together with observations of altered Fz pathway protein localization in Ds-Fat pathway mutants, led to suggestions that the Ds-Fat pathway acts upstream of the Fz pathway (*Ma et al., 2003*; *Yang et al., 2002*). According to this hypothesis, the Ds-Fat pathway acts as a 'global module' that provides long-range directional information through tissue-wide Fj and Ds gradients, whereas the Fz pathway acts as a 'core module'

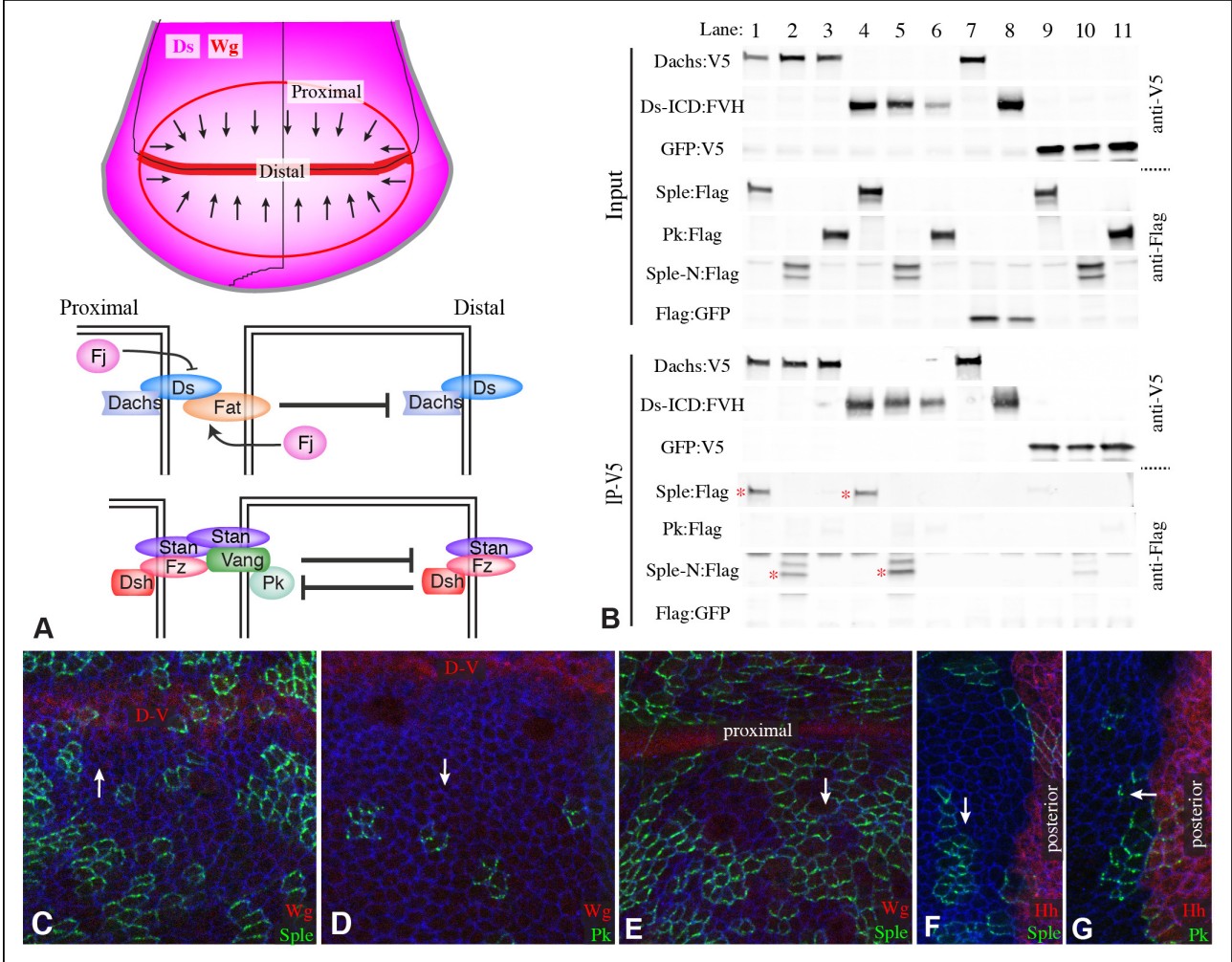

**Figure 1.** Localization of Pk and Sple in wing discs, and their interaction with Dachs and Ds. (**A**) Schematic diagram illustrating the general direction of PCP protein polarity (arrows), expression gradient of Ds (magenta) and organization of Ds-Fat and Fz PCP pathway components in the *Drosophila* wing disc. (**B**) Western blots, using antibodies indicated on the right, showing the results of co-immunoprecipitation experiments between V5-tagged Dachs (lanes 1–3,7), Ds-ICD (lanes 4–6,8) or GFP (lanes 9–11) of Flag-tagged Sple (lanes 1,4,9), Sple-N (lanes 2,5,10), Pk (lanes 3,6,11) or GFP (lanes 7,8). Upper panels (Input) show blots on lysates of S2 cells, lower panels (IP-V5) show blots on proteins precipitated from these lysates by anti-V5 beads. Similar results were obtained in three independent biological replicates of this experiment. (**C–G**) Portions of wing imaginal discs with clones of cells expressing GFP:Sple (**C,E,F**) or GFP:Pk (**D,G**) (green), stained for expression of E-cadherin (blue), and showing either anti-Wg (**C–E**) or hh-Gal4 UAS-mCD8-RFP (**F,G**) (red). White arrows indicate direction of polarization of Sple or Pk.

The following figure supplements are available for figure 1:

**Figure supplement 1.** Proteins used in co-immunoprecipitation assays.

**Figure supplement 2.** Ds and Fj gradients in wing discs.

that establishes robust polarization that can propagate locally, and effects cellular polarity. This suggestion was challenged by observations that clones of cells mutant for or over-expressing *ds, fj* or *fat* in the abdomen can affect PCP non-autonomously even in the absence of Fz pathway components (*Casal et al., 2006*). Additionally, in the abdomen, combining mutations in both Ds-Fat and Fz pathway genes can have more severe effects on PCP than single mutants, suggesting that these pathways can act in parallel (*Casal et al., 2006*; *Donoughe and DiNardo, 2011*; *Repiso et al., 2010*). There are also some manifestations of PCP, such as oriented cell divisions in the developing

wing, which are influenced by the Ds-Fat pathway and not the Fz pathway (*Baena-Lopez et al., 2005*).

Nonetheless, other studies have provided evidence of cross-talk between PCP systems, and implicated the Pk-Sple locus in helping to mediate this cross-talk. The Pk-Sple locus produces two functional isoforms: Prickle (Pk) and Spiny-legs (Sple), which share a common, LIM-domain containing C-terminus, but unique N-termini (*Figure 1—figure supplement 1*) (*Gubb et al., 1999*). These isoforms have distinct roles: for example, mutations that only affect *pk* disrupt PCP in the wing and notum, but not in the eye and leg, whereas mutations that only affect *sple* exhibit a complementary specificity. The observations that mutations that affect both isoforms (*pk-sple*) have milder effects on PCP than isoform-specific alleles in the wing, notum, and leg, and that over-expression of Sple or Pk leads to PCP phenotypes reminiscent of loss-of-function of *pk*, or *sple*, respectively, led to the suggestion that a balance between Pk and Sple isoforms is necessary for normal PCP (*Gubb et al., 1999*). Studies of PCP establishment in the pupal wing revealed that it occurs in distinct phases, and suggested that influences of Sple are correlated with influences of the Ds-Fat pathway (*Hogan et al., 2011*; *Merkel et al., 2014*). Moreover, examination of PCP protein localization revealed that a coupling between the polarization of components of the Ds-Fat and Fz systems could be induced by expression of Sple (*Merkel et al., 2014*). Additionally, in the abdomen, the Fj and Ds gradients are oriented oppositely within anterior (A) versus posterior (P) compartments of each segment (*Casal et al., 2002*). Since hairs always point posteriorly, this led to the suggestion that there could exist a 'rectification' mechanism, which would reverse the influence of these gradients on hair polarity. The observations that Sple over-expression reverses polarity in P compartments, and that *pk-sple* mutants reverse polarity in part of the A compartment, led to the suggestion that Pk and Sple might be involved in this rectification (*Lawrence et al., 2004*).

Two potential mechanisms by which Pk-Sple might influence the relationship between PCP pathways have recently been suggested. It was reported that Dachs could directly interact with Pk and Sple, and that Ds and Dachs could influence Sple localization in wing discs (*Ayukawa et al., 2014*). It has also been proposed that Pk-Sple could connect PCP pathways through an influence on microtubule orientation (*Olofsson et al., 2014*). Vesicles containing Fz and Dsh have been observed to move along apical non-centrosomal microtubules towards the distal side of wing cells, with the proximal-distal alignment of microtubules and consequent directional transport of Fz pathway components dependent upon the Ds-Fat pathway (*Harumoto et al., 2010*; *Matis et al., 2014*; *Olofsson et al., 2014*; *Shimada et al., 2006*). Pk and Sple also influence the orientation of apical microtubules, such that the plus ends of microtubules are preferentially found at either the high end or the low end of the Ds gradient, depending on whether Pk or Sple, respectively, is the predominant isoform (*Matis et al., 2014*; *Olofsson et al., 2014*). Relative differences in expression of isoforms consistent with their distinct requirements have also been reported: Pk at higher levels than Sple in larval wing discs, and Sple at higher levels than Pk in eye discs (*Ayukawa et al., 2014*; *Merkel et al., 2014*; *Olofsson et al., 2014*). While these studies are suggestive of a key role for Pk-Sple in linking PCP pathways, the extent to which these or other mechanisms link PCP pathways, and their contribution to orienting PCP, remain unclear.

Here, we demonstrate that Dachs and Ds can each physically interact with Sple, and control its localization in the wing, eye and abdomen. Our studies complement observations of *Ayukawa et al. (2014)* in identifying requirements for Dachs and Ds in Sple localization, but differ regarding the nature of these requirements. We also extend understanding of the relationship between Ds-Fat and Fz PCP pathways by identifying organ and region-specific differences in their interactions, and illustrate how this relationship between pathways can explain poorly understood features of PCP mutant phenotypes. Our results establish control of Sple localization as a key mechanism by which the Ds-Fat pathway coordinates with Fz to influence PCP, and enhance our understanding of how PCP is coordinated in developing tissues.

## Results

### Distinct localization of Pk and Sple in wing imaginal discs

Components of each of the two major PCP pathways are polarized along the proximal-distal axis of the larval wing imaginal disc and pupal wing (*Figure 1A*) (*Goodrich and Strutt, 2011*; *Matis and*

*Axelrod, 2013*). Components of both pathways are required for the normal distal orientation of wing hairs. Motivated by observations implicating the *pk-sple* locus in modulating the influence of the Ds-Fat pathway on wing hair and ridge polarity (*Hogan et al., 2011*), we initiated experiments to examine the localization of the distinct Pk and Sple isoforms and their potential regulation by Ds-Fat PCP. This was achieved by expressing GFP-tagged isoforms in clones of cells. Consistent with recent studies examining isoform-specific localization (*Ayukawa et al., 2014*; *Sagner et al., 2012*; *Strutt et al., 2013*), we observed that GFP:Pk was polarized towards the proximal sides of cells, except just anterior to the anterior-posterior compartment boundary, where GFP:Pk was instead polarized towards the anterior sides of cells (*Figure 1D,G*, *2I*). By contrast, GFP:Sple was polarized towards the distal sides of cells throughout the wing disc (*Figure 1C,E,F*, *2I*). The distinct localization of Pk and Sple expressed in wing discs indicates that they can respond to distinct spatial cues.

## Dachs and Ds can physically interact with Sple

The localization of Sple to the distal side of wing disc cells is similar to that of Dachs and Ds (*Figure 1—figure supplement 2*) (*Ambegaonkar et al., 2012*; *Brittle et al., 2012*; *Mao et al., 2006*; *Rogulja et al., 2008*). To investigate whether this shared localization could reflect physical association, we assayed for co-immunoprecipitation of epitope-tagged proteins expressed in cultured *Drosophila* S2 cells (*Figure 1—figure supplement 1*). Indeed, V5-tagged Dachs could co-immunoprecipitate Flag-tagged Sple (*Figure 1B*, lane 1). Dachs and Sple interact through the unique N-terminus of Sple, because Dachs also co-precipitated a construct comprising only the Sple N-terminus (Sple-N), but did not co-precipitate a full length Pk construct (*Figure 1B*, lanes 2 and 3). Interaction with Ds was investigated by expressing a construct comprising the entire intracellular domain of Ds (Ds-ICD). Both Sple and Sple-N also interacted with Ds-ICD, whereas Pk did not (*Figure 1B*, lanes 4–6). We note that *Ayukawa et al. (2014)* similarly reported an ability of Dachs to interact with Sple, based on co-immunoprecipitation of proteins expressed in human HEK293 cells. However, our results differ in that they reported that Dachs could also interact with Pk, whereas we could not detect any interaction between Pk and Dachs above non-specific background (defined by precipitation observed using GFP:V5 instead of Dachs:V5, *Figure 1B*, lane 11). Also, *Ayukawa et al. (2014)* reported that they could detect an interaction between Sple-N and Ds-ICD, but could not detect an interaction between Ds-ICD and full length Sple, leading them to suggest a requirement for other components such as Dachs, whereas we did detect this interaction (*Figure 1B*, lane 4). Altogether, our results establish that Dachs and Ds can each independently interact with Sple, and that they do so through its unique N-terminal region.

## Influence of Dachs and Ds on Sple localization in wing discs

To determine whether the shared distal localization and physical interaction between Dachs and Sple are reflective of a functional role for Dachs in localizing Sple, we examined GFP:Sple in *dachs* mutant wing discs. Indeed, GFP:Sple localization was altered, as throughout most of the developing wing disc its localization became similar to that of Pk: on the proximal side of cells, and in fewer, more discrete puncta (*Figure 2A,I*). Along the A-P compartment boundary, GFP:Sple was instead localized towards anterior side of cells, as is GFP:Pk (*Figures 2I*, *3A*) (*Sagner et al., 2012*). Intriguingly, however, in the most proximal part of the wing pouch, GFP:Sple generally maintained a distal localization (*Figure 2B,I*). Thus, Dachs is required for the distal localization of GFP:Sple throughout most of the wing pouch, but not in the proximal wing.

In *ds* mutant wing discs, GFP:Sple was in most cases unpolarized (localized to cell membrane on all sides), but sometimes partially polarized (on multiple cell sides but with a clear bias), or polarized in random directions (*Figure 2D,I*). The observation of unpolarized GFP:Sple is consistent with the inference that Dachs can localize GFP:Sple, because Dachs is localized to all membranes in an unpolarized fashion in *ds* mutants (*Ambegaonkar et al., 2012*). Dachs is similarly localized to the membrane in an unpolarized fashion in *fat* mutants (*Mao et al., 2006*), and we always observed unpolarized membrane localization of GFP:Sple in *fat* mutant wing discs (*Figure 2C*). To confirm that mis-localization of Dachs is responsible for the mis-localization of GFP:Sple in *ds* or *fat* mutant wing discs, we examined GFP:Sple in *ds dachs* and *fat dachs* double mutants. In both cases, GFP:Sple reverted to a Pk-like localization, including a proximal, punctate orientation throughout most of the wing pouch, and an anterior orientation near the A-P boundary (*Figure 2E–I*).

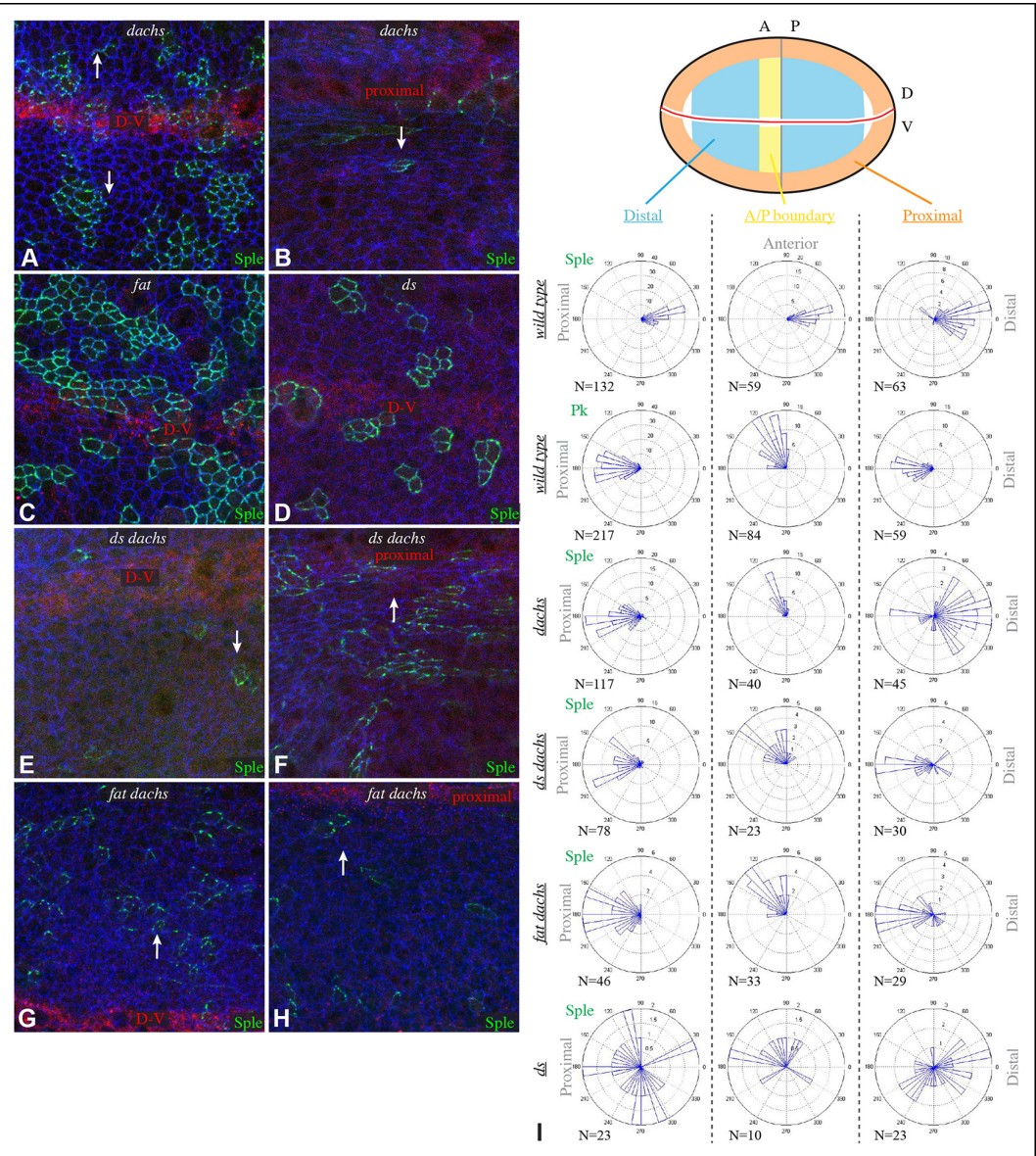

**Figure 2.** Localization of Sple in Ds-Fat pathway mutants. (A–H) Portions of wing imaginal discs with clones of cells expressing GFP:Sple (green) in $d^{GC13}/d^{210}$ (A and B), $ft^8/ft^{G-rv}$ (C), $ds^{36D}/ds^{UA071}$ (D), $d^{GC13}ds^{36D}/d^{GC13}ds^{UA071}$ (E and F) and $d^{GC13}ft^8/d^{GC13}ft^{G-rv}$ (G and H) mutants. Discs were stained for E-cad (blue) and Wg (red). Wg is expressed along the D-V boundary and in proximal rings, the locations of Wg expression shown are indicated. White arrows indicate general direction of Sple polarization. (I) Rose plots summarizing orientation of Sple or Pk in the indicated genotypes, with proximal at left, distal at right, and in the central row, anterior at top. Orientations were scored separately in the three regions depicted in the cartoon, the number of cells scored is indicated by N.

One remarkable feature of *ds dachs* or *fat dachs* mutant discs is that GFP:Sple localization is preferentially proximal even in proximal regions of the wing pouch, where GFP:Sple localization was preferentially distal in *dachs* mutants (*Figure 2B,F,H,I*). This implies that the distal localization of GFP:Sple here in *dachs* mutants is Ds-dependent, and hence that Dachs and Ds each have the ability to independently localize GFP:Sple. Thus, in proximal cells, where Ds expression is higher, Dachs and Ds could provide redundant localization cues for GFP:Sple. In distal cells, by contrast, we suggest that Dachs could be required for GFP:Sple localization because Ds expression is too low. In *ds* or *fat* mutants, Dachs is mis-localized, and Ds is either absent (in *ds* mutants) or unpolarized with reduced junctional accumulation (in *fat* mutants) (*Ma et al., 2003*; *Mao et al., 2009*; *Strutt and*

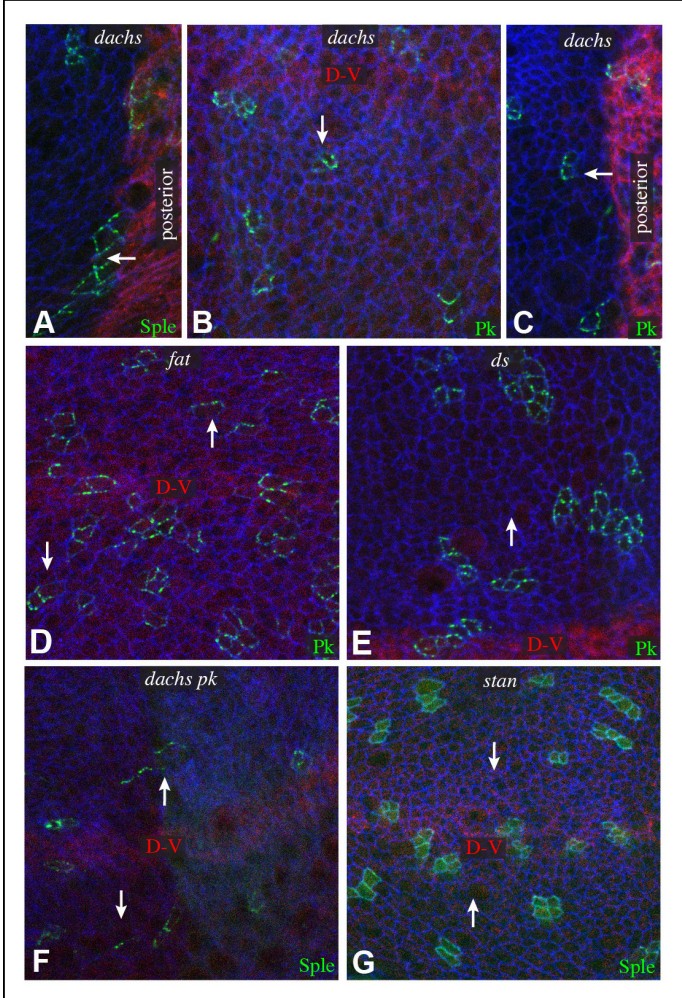

**Figure 3.** Additional characterization of Pk and Sple localization in mutants. Portions of wing discs with clones of cells expressing GFP:Sple (**A,F,G**) or GFP:Pk (**B–E**) (green) in $d^{GC13}/d^{210}$ (**A–C**), $ft^8/ft^{G-rv}$ (**D**), $ds^{36D}/ds^{UA071}$ (**E**), $d^{GC13} pk^{30}$ (**F**), and $vang^{stbm6}$(**G**) mutants. Discs were stained for E-cad (blue) and Wg (red) (**B,D,E,F** and **G**) or hh-Gal4 UAS-mCD8-RFP (red) (**A,C**). The white arrows indicate direction of polarization.

*Strutt, 2002*), and consequently GFP:Sple becomes mis-localized. Finally, in the absence of both Dachs and Ds localization cues, as in *fat dachs* or *ds dachs* mutants, GFP:Sple follows Pk localization cues. In principle this could occur either because Sple is able to respond directly to the same cues as Pk through shared motifs, or because Sple can bind to Pk. The observation that GFP:Sple localized proximally in *dachs pk* mutant wing discs (*Figure 3F*) indicates that Sple can respond to Pk localization cues even in the absence of Pk. Localization of Pk was not visibly altered within larval wing discs by *dachs, ds,* or *fat* mutations (*Figure 3*). Fz-PCP is not required for distal localization of Sple, as GFP:Sple remains preferentially distal in *vang* mutant wing discs (*Figure 3G*). Cytoplasmic levels of Sple were also visibly increased in *vang* mutants, consistent with studies of the influence of Vang on Pk levels (*Strutt et al., 2013*).

We note that our studies agree with *Ayukawa et al. (2014)* in reporting an influence of *dachs* and *ds* on Sple localization, but differ in that they reported that in *ds* mutants Sple polarity was reversed, resembling Pk, whereas we observe either a complete absence, or a randomization, of Sple polarization in *ds* mutants, consistent with Sple being regulated by Dachs. Also, they did not report observing the difference in localization of Sple between distal and proximal regions of *dachs* mutants, which we determined reflects a Dachs-independent regulation of Sple by Ds. Nonetheless,

our studies agree that a direct connection between the Fz and Ds-Fat PCP pathways can be mediated through Dachs and Ds-dependent control of Sple.

## PCP in *pk* mutant wings reflects Dachs-directed polarity

The link between PCP pathways mediated through Sple has important implications for how PCP is oriented, and suggests explanations for the basis of both *pk* and *fat* mutant polarity phenotypes. *sple* mutation does not result in a hair polarity phenotype in the wing, whereas *pk* mutants have a characteristic wing polarity phenotype, in which hairs in much of the wing are mis-oriented away from the wing margin, and wing margin bristles can point proximally (*Figure 4D,H*; *Figure 4—figure supplement 1*) (*Gubb et al., 1999*). The observation that Pk and Sple can localize in opposite directions, whereas wing hairs normally point in a single direction, implies that cells must ultimately choose which of these two distinct localization cues to follow. Normally, they choose the Pk cue (*Figure 4A*), presumably because the Pk isoform is more abundant than the Sple isoform in the wing (*Ayukawa et al., 2014*; *Merkel et al., 2014*; *Olofsson et al., 2014*), and hence it dictates polarization. Indeed, if Sple is over-expressed, then hair polarity is reversed even more strongly than in *pk* mutants, and can align with Ds-Fat PCP (*Ayukawa et al., 2014*; *Doyle et al., 2008*; *Gubb et al., 1999*; *Merkel et al., 2014*; *Olofsson et al., 2014*), and the expression of GFP:Sple in clones is sufficient to alter wing hair polarity (*Figure 4—figure supplement 1*). These observations suggest that wing hair polarity in *pk* mutants could be directed by the Ds-Fat pathway dependent polarization of Sple (*Figure 4A*). As this linkage in most of the wing depends upon *dachs* (*Figure 2I*), this hypothesis predicts that the *pk* wing hair polarity phenotype should be suppressed by *dachs* mutation (*Figure 4A*). Indeed, when we tested this by comparing wing hair and bristle orientation in *pk* versus *dachs pk* mutants, this suppression was observed (*Figure 4C–J*). This result can also explain the observation that over-expression of Fat could suppress the *pk* hair polarity phenotype (*Hogan et al., 2011*), because over-expression of Fat removes Dachs from the membrane (*Mao et al., 2006*), which, as Dachs functions at membranes (*Pan et al., 2013*; *Rauskolb et al., 2011*), is functionally equivalent to *dachs* mutation.

## Sple contributes to Fat PCP phenotypes in the wing

The determination that Ds-Fat and Fz pathways are molecularly linked by physical interaction between Dachs and Sple also provides a new perspective on polarity phenotypes of *fat* and *ds*. The altered wing hair polarity in *fat* or *ds* mutants has been interpreted as indicating that Fat and Ds have a normal role in directing hair polarity in regions of the wing. Indeed, recent studies have inferred that Ds-Fat PCP influences core protein polarization in the wing by orienting microtubules (*Harumoto et al., 2010*; *Matis et al., 2014*; *Olofsson et al., 2014*). However, as hair polarity in the wing is normally Pk-dependent rather than Sple-dependent, and as we found that Ds-Fat PCP in the wing influences Sple localization but not Pk localization, we considered an alternative model: rather than reflecting a normal role in directing hair polarity, these phenotypes of *fat* and *ds* could stem from the inappropriate accumulation of Dachs, leading to inappropriate localization of Sple, which in some contexts could interfere with the normal Pk-dependent polarization cues (*Figure 4A*). Consistent with this hypothesis, *ds* hair polarity phenotypes are suppressed by *dachs* (*Brittle et al., 2012*), and we confirmed that *fat* wing hair polarity phenotypes (generated using wing-specific RNAi) are also suppressed by *dachs* (*Figure 4—figure supplement 1*). This hypothesis further predicts that *fat* PCP phenotypes could be suppressed by *sple* (*Figure 4A*), and while we did not observe completely normal hair polarity in *fat sple* wings, we did observe a partial suppression, including restoration of normal, distally oriented polarity in two regions affected by loss of Fat: near the proximal anterior wing margin, and near the anterior cross-vein (*Figures 4B,K–N*; *Figure 4—figure supplement 1*). By contrast, when Hippo pathway phenotypes of *fat* are rescued by Warts over-expression, wing polarity remains abnormal in the proximal wing (*Feng and Irvine, 2007*) (*Figure 4—figure supplement 1*). Thus, while Sple is not required for normal wing hair polarity, it mediates a connection between Ds-Fat and Fz pathways that contributes to abnormal hair polarity in the absence of *fat*.

## Control of Sple polarity in eye discs by Ds-Fat PCP

PCP in the eye has been studied for its influence on the organization and orientation of ommatidia (*Jenny, 2010*). The eight photoreceptor cells within each ommatidia are arranged in a characteristic

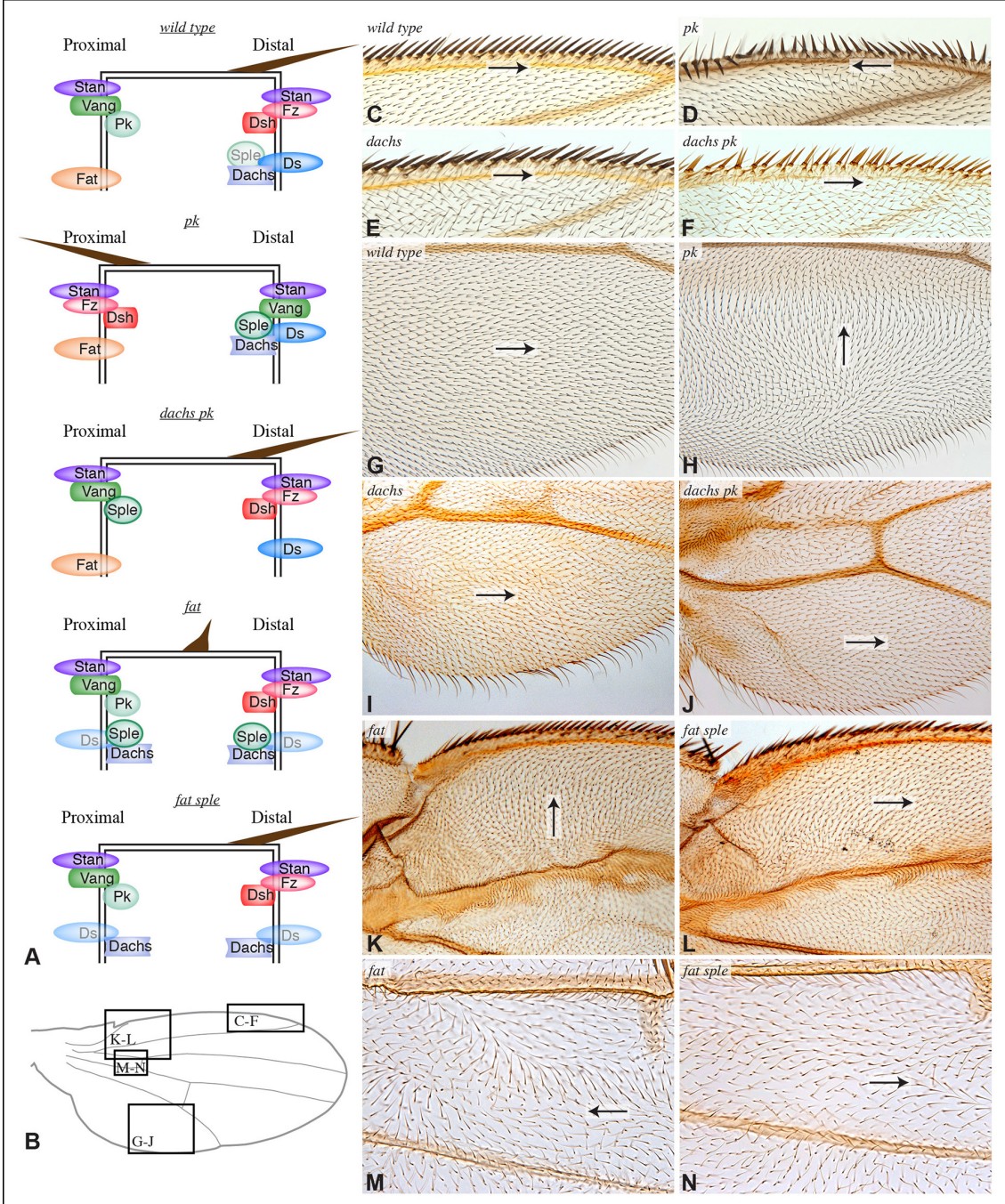

**Figure 4.** Contribution of Dachs and Sple to PCP mutant wing phenotypes. (**A**) Cartoons depicting inferred protein localization and hair orientation (brown) in wing cells of the indicated genotypes to explain rescue of *pk* by *dachs*, and rescue of *fat* by *sple*. Faint Sple and Ds indicate lower levels. (**B**) Schematic adult wing to show approximate location of panels shown in close-up, as indicate by letters. (**C–N**) Close-ups of portions of wings (as indicated in panel **B**) to show hair and bristle orientation in the indicated genotypes. Arrows indicate general direction of polarity. (**C–F**) Show wing margin bristles, (**G–N**) show wing hairs, in wild type (**C,G**), $pk^{30}$ (**D,H**), $d^{GC13}/d^{210}$ (**E,I**), $d^{GC13}$ $pk^{30}$ (**F,J**), *UAS-RNAi-fat/+; C765-Gal4/UAS-dcr2* (**K,M**) and $sple^1$ *UAS-RNAi-fat/ $sple^1$; C765-Gal4/UAS-dcr2* (**L,N**). Suppression of *pk* polarity phenotypes by *dachs* was 100% penetrant. For suppression of *fat* phenotypes by *sple*, near the proximal anterior wing margin (region K,L) in 10/10 *fat* RNAi wings scored hairs point predominantly towards the wing margin, whereas in 7/8 *fat sple* wings scored hairs point predominantly distally, and in 1/8 wings scored a substantial fraction of hairs (~1/4) point towards the wing margin. Near the anterior cross-vein (region M-N), in 8/9 *fat* RNAi wings scored hairs point predominantly proximally, and in 1/9 wings scored hairs point predominantly towards the L3 vein, whereas in 5/8 *fat sple* wings scored hairs point distally, in 2/8 they point towards the L3 vein, and in 1/8 they point proximally.

*Figure 4 continued on next page*

*Figure 4 continued*

The following figure supplement is available for figure 4:

**Figure supplement 1.** Hair polarity in wing is not affected by loss of *sple* or *dachs*.

pattern that comes in two chiral forms. This chirality is determined by which of two neighboring photoreceptors becomes the R3 cell and which becomes R4. This decision is dependent upon Notch signaling, which is biased by Fz PCP such that the cell at the R3-R4 interface with higher Fz becomes R3 (*Cooper and Bray, 1999*; *Fanto and Mlodzik, 1999*; *Strutt et al., 2002*; *Tomlinson and Struhl, 1999*). The two chiral forms are established in mirror symmetry with respect to the dorsal-ventral compartment boundary, termed the equator (*Figure 5—figure supplement 1*). In *sple* mutants, ommatidial chirality is randomized, whereas in *pk* mutant eyes ommatidial chirality is normal (*Gubb et al., 1999*). Ds and Fj are expressed in complementary gradients in the eye (*Figure 5—figure supplement 1*), and experiments manipulating Ds and Fj expression have revealed that these gradients instruct normal polarity (*Simon, 2004*; *Strutt and Strutt, 2002*; *Yang et al., 2002*; *Zeidler et al., 1999*). However, the relationship between Fz and Ds-Fat PCP pathways in the eye and how this influences polarity has remained unclear. Also, in contrast to the wing and abdomen, where *dachs* mutation suppresses *fat* PCP phenotypes, *dachs* mutation has little effect on *fat* PCP phenotypes in the eye (*Brittle et al., 2012*; *Mao et al., 2006*; *Sharma and McNeill, 2013*). We hypothesized that the influence of Ds-Fat PCP on ommatidial polarity might be accounted for by an ability of Ds to polarize Sple independently of *dachs*, as in the proximal wing.

Ommatidia form progressively in a wave of differentiation that sweeps across the eye disc, initiated within a line of cells that form the morphogenetic furrow. We analyzed GFP:Sple localization at the 5-cell precluster stage of ommatidial formation, when R3-R4 specification occurs. GFP:Sple localized to the equatorial side of cells within both R3 and R4, which places it within R4 at the R3-R4 interface (*Figure 5A*). Equatorial polarization of Sple co-localizes it with Vang (*Strutt, 2001*), and is consistent with the observation that it interacts with Ds, since Ds is also polarized to the equatorial side of cells in the eye disc (*Brittle et al., 2012*). This equatorial polarization of Sple was disrupted in *ds* or *fat* mutants (*Figure 5C,E*), but was not affected by mutation of *dachs* (*Figure 5B*), nor could *dachs* mutation prevent the mis-localization of Sple in *ds* or *fat* (*Figure 5D,F*). In *ds* or *fat* mutants, Sple localization was partially randomized within R3 and R4, and also partially unpolarized, in that it was often detected on multiple cell junctions. However, it was never detected along the cell junction with the more anterior cells within the ommatidial cluster (R2 and R5) (*Figure 5*).

We also examined localization of GFP:Pk within R3 and R4, and found that it too localized to the equatorial side of both cells (*Figure 5G*). How might Pk localize equatorially if it cannot interact with Ds? We hypothesized that this might arise from an ability of Pk-Sple proteins to multimerize, or from the interactions that lead core components of the Fz pathway to adopt a shared, discrete, polarized localization. In such cases, equatorial polarization of Pk would not come about because it directly responds to an equatorial-polar signal like Ds, but rather because it can interact with Sple and/or Vang, which recruit it to equatorial cell junctions. In support of this, we found that in *sple* mutants, Pk localization is altered such that it becomes partially randomized, resembling Sple localization in *fat* or *ds* mutants (*Figure 5H*). Thus, equatorial Pk localization in R3 and R4 depends upon Sple.

The posterior bias in Pk localization in *sple* mutants, and the similar posterior bias in Sple localization within *ds* or *fat* mutants, tend to place Pk or Sple towards the side of the cell nearest the morphogenetic furrow (*Figure 5*). The morphogenetic furrow is a source of local signals for multiple pathways, including Notch, Hedgehog and Decapentaplegic, which might, indirectly at least, influence Fz PCP orientation in these mutant eye discs, as they have been implicated in influencing PCP in other contexts (*Sagner et al., 2012*; *Struhl et al., 1997*). Finally, we observed that in front of the morphogenetic furrow, GFP:Sple and GFP:Pk exhibit distinct localization profiles, with GFP:Sple accumulating on the equatorial sides of cells, as it does behind the furrow, but Pk:GFP accumulating on the anterior sides of cells, which in this region is the side closest to the morphogenetic furrow (*Figure 5—figure supplement 2*). Thus, behind the morphogenetic furrow, Pk and Sple co-localize in an Sple-dependent process, whereas in front of the morphogenetic furrow, Sple and Pk can localize differently.

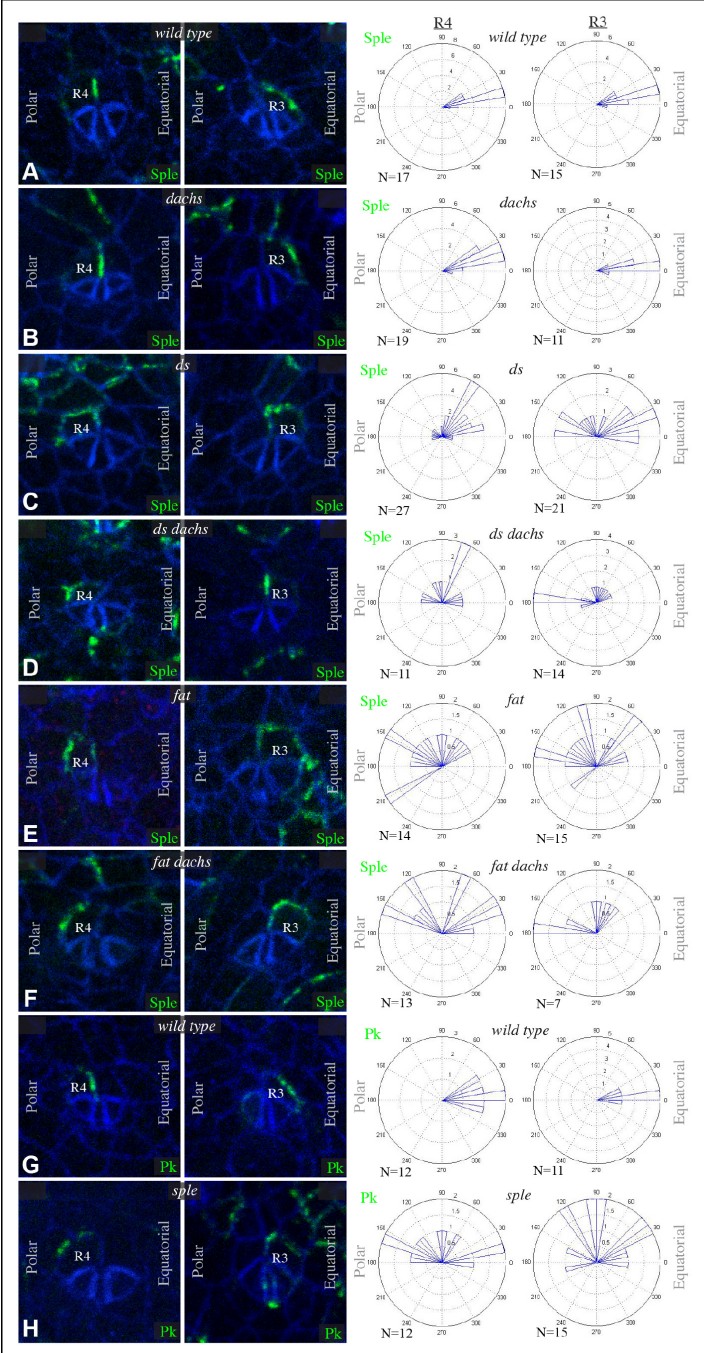

**Figure 5.** Sple and Pk localization in photoreceptor cells. Localization of GFP:Sple (**A–F**) or GFP:Pk (**G,H**) in cells with expression in R4 or R3 photoreceptor cells in wild type (**A,G**), $d^{GC13}/d^{210}$ (**B**), $ds^{36D}/ds^{UA071}$ (**C**), $d^{GC13}\ ds^{36D}/d^{GC13}\ ds^{UA071}$ (**D**), $ft^8/ft^{G-rv}$ (**E**), $d^{GC13}\ ft^8/d^{GC13}\ ft^{G-rv}$ (**F**), and $sple^1$ (**H**) mutants. Rose plots summarize localization based on the indicated number (**N**) of examples, with equatorial to the right, polar to the left, and anterior (towards the morphogenetic furrow) at top.

The following figure supplements are available for figure 5:

**Figure supplement 1.** Polarity and gradients in eye discs.

**Figure supplement 2.** Sple and Pk polarity in front of the morphogenetic furrow, and influence of GFP:Pk on PCP.

Our observation that GFP:Pk localizes, like Sple, on the equatorial sides of R3 and R4 behind the furrow was initially unexpected, because abnormal PCP has been reported in adult eyes with clones of cells expressing GFP:Pk (*Strutt et al., 2013*). However, the observation of a distinct localization for GFP:Pk in front of the furrow suggested that the timing of GFP:Pk expression might be important. Polarization of GFP:Pk or GFP:Sple is examined in clones induced 1 day before dissection, as it is scored in ommatidia in which these transgenes are expressed only in R3 or only in R4. These clones are induced when cells are in or behind the furrow. When we examined PCP in eye discs, using Prospero and Elav to stain ommatidia, PCP appeared unaffected by clones expressing GFP:Pk for one day (*Figure 5—figure supplement 2*). By contrast, within 3-day-old clones, which would have been initiated in front of the morphogenetic furrow, PCP was disturbed (*Figure 5—figure supplement 2*). These observations indicate that Pk over-expression can disturb PCP in the eye, but PCP ultimately becomes refractory to Pk over-expression.

## Interactions between PCP pathways in the abdomen

Hairs in the *Drosophila* abdomen point posteriorly; this orientation is influenced by components of both the Fz and Ds-Fat PCP pathways (*Casal et al., 2002*; *2006*; *Lawrence et al., 2004*). In analyzing the relationship between PCP pathways in the abdomen, we focused on the pleural cells, which form in lateral and ventral regions, but have also examined polarity in tergites, which form on the dorsal side of the abdomen. As the subcellular localizations of Dachs, Sple and Pk within pupal abdominal cells have not been described, we first characterized their distributions in pleural cells of wild-type animals at pupal stages, with posterior compartments marked using *hh-Gal4* and *UAS-RFP* transgenes. Dachs:GFP and Sple:GFP were polarized towards the anterior sides of cells within A compartments, and towards the posterior sides of cells within P compartments (*Figure 6A,B,H*). This is consistent with their being polarized in response to the Ds and Fj gradients, as the Fj and Ds gradients are oriented oppositely within anterior (A) versus posterior (P) compartments of each segment (*Figure 6—figure supplement 1*) (*Casal et al., 2002*), and Dachs and Sple accumulate on the sides of cells that face towards lower Ds levels and higher Fj levels. Pk:GFP, by contrast, was polarized towards the anterior sides of cells within both A and P compartments (*Figure 6C,H*). Thus, in A compartments Pk:GFP and Sple:GFP polarize in the same direction, whereas in P compartments they polarize in opposite directions.

Consistent with prior studies (*Lawrence et al., 2004*), we observed that *pk-sple* mutants reverse hair polarity within the center of the A compartment, while the P compartment, and the edges of the A compartment, exhibit normal hair polarity (*Figure 7B,F*). The A compartment in tergites encompasses all of the hairs and bristles in the anterior, pigmented part of each abdominal segment, plus approximately two rows of hairs in the unpigmented region posterior to this (*Struhl et al., 1997*). The P compartment in tergites encompasses the remaining hairs posterior to the A compartment, plus a region of naked cuticle. In pleura we estimated the A and P compartment regions based on the neighboring tergites, but can not make precise assignments of compartment identity for hairs near compartment boundaries. We extended analysis of *pk-sple* by examining isoform-specific alleles. *pk* mutant alleles have normal polarity in A compartments, but mostly reversed polarity in the P compartment within pleura (*Figure 7C*), although not in tergites (*Figure 7G*). *sple* mutant alleles have normal polarity in the P compartment, and abnormal polarity, including hair reversal but also sideways or swirling hair orientations, within the center of the A compartment (*Figure 7D,H*).

The influence of *pk-sple* on polarity in the P compartment of the pleura thus appears reminiscent of the situation in the wing: Pk and Sple can respond to opposing localization cues. In the absence of Pk, cells respond instead to Sple-dependent cues, leading to a reversal of polarity (Sple localized normally in *pk* mutants, *Figure 8I,K*). The essential contribution of Sple to the *pk* phenotype is confirmed by its suppression in *pk-sple* mutants (*Figure 7*). The influence of Pk-Sple on polarity in A compartments is reminiscent of the situation in the eye. Sple, not Pk plays the key role in establishing polarity here, and Pk was mis-localized within the central region of A compartments of *sple* mutants (*Figure 6G,H*), hence Sple contributes to localization of Pk here.

In *fat* or *ds* mutants, hair polarity is disturbed in much of the A and P compartments, although a small region at the front of the A compartment exhibits normal hair polarity (*Figures 9A,E*, *10A, E*) (*Casal et al., 2002*). To determine whether the abnormal polarity could be explained by mis-localization of Dachs and/or Ds, and a consequent mis-localization of Sple and/or Pk, we assessed both

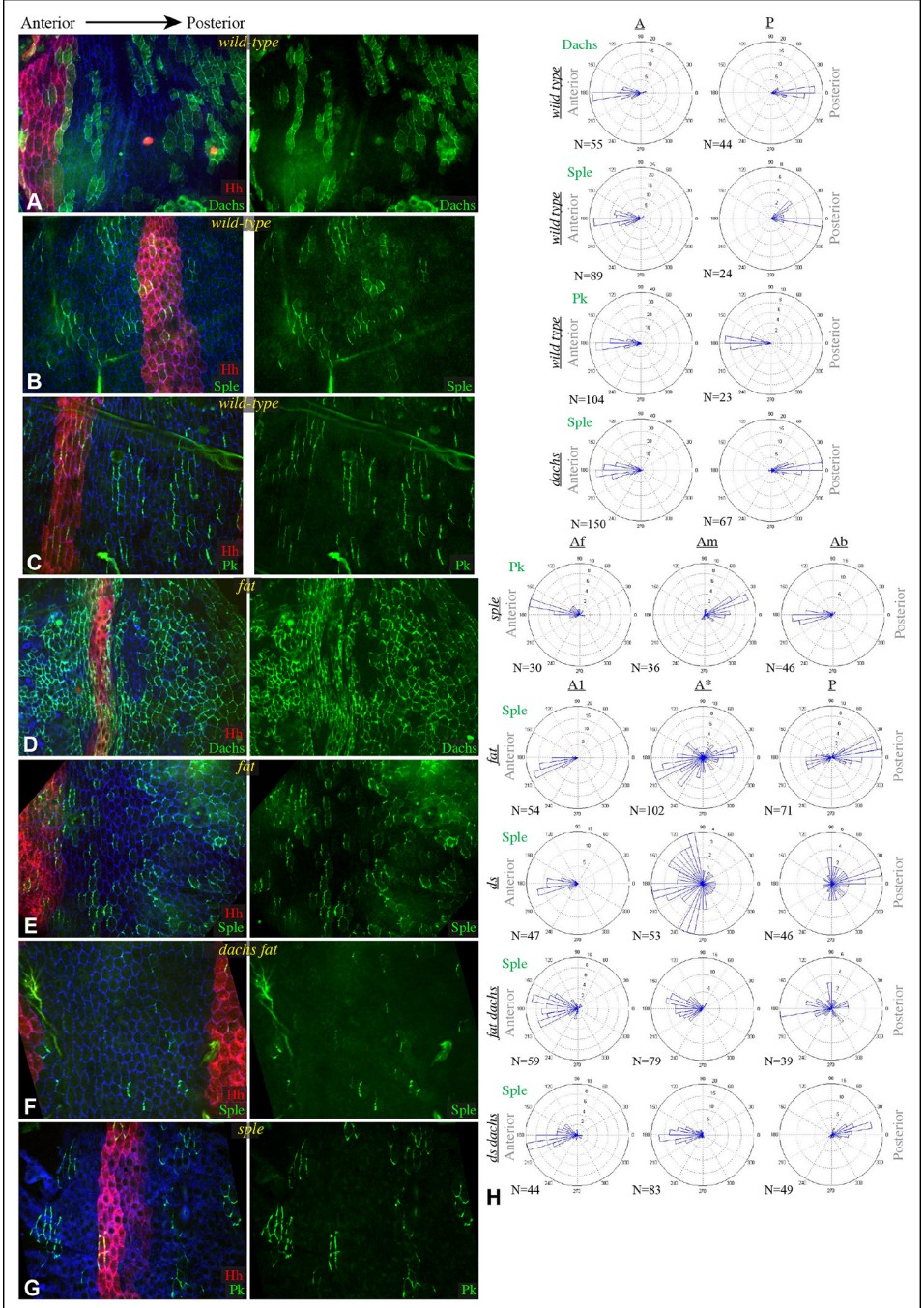

**Figure 6.** Localization of Dachs, Sple and Pk in abdominal pleura. (A–G) Pleura of wildtype (A–C), $ft^8/ft^{G-rv}$ (D,E), $d^{GC13} ft^8/d^{GC13} ft^{G-rv}$ (F) and $sple^1/sple^1$ (G) pupae with clones of cells expressing GFP:Dachs (A,D), GFP:Sple (B,E,F) and GFP:Pk (C,G) (green). Posterior compartments are marked by hh-Gal4 UAS-mCD8-RFP (red). Anterior-posterior body axis is indicated at top. (H) Rose plots depicting polarization of GFP:Dachs, GFP:Sple or GFP:Pk in pleural cells of the indicated genotypes; anterior polarization is to the left and posterior polarization is to the right. For wild type and *dachs* mutants cells were scored separately in A and P compartments. For *fat*, *ds*, *fat dachs*, and *ds dachs* the anterior compartment was further subdivided into a front region (A1, anterior-most 8 cells), and the remainder of the A compartment (A*). For *sple*, the A compartment was subdivided into a front region of 5 cells (Af), a back region of 10 cells (Ab), and a middle region comprising the rest of the compartment (Am); P compartment localization is summarized in *Figure 6—figure supplement 1*. Scoring of sub-regions of the A compartment in wild-type is shown in *Figure 6—figure supplement 1*. Apparent variations in cell size are mostly due to the flexibility of the pleura, which is easily stretched or compressed.

The following figure supplement is available for figure 6:

**Figure supplement 1.** Gradients influencing PCP in the abdomen.

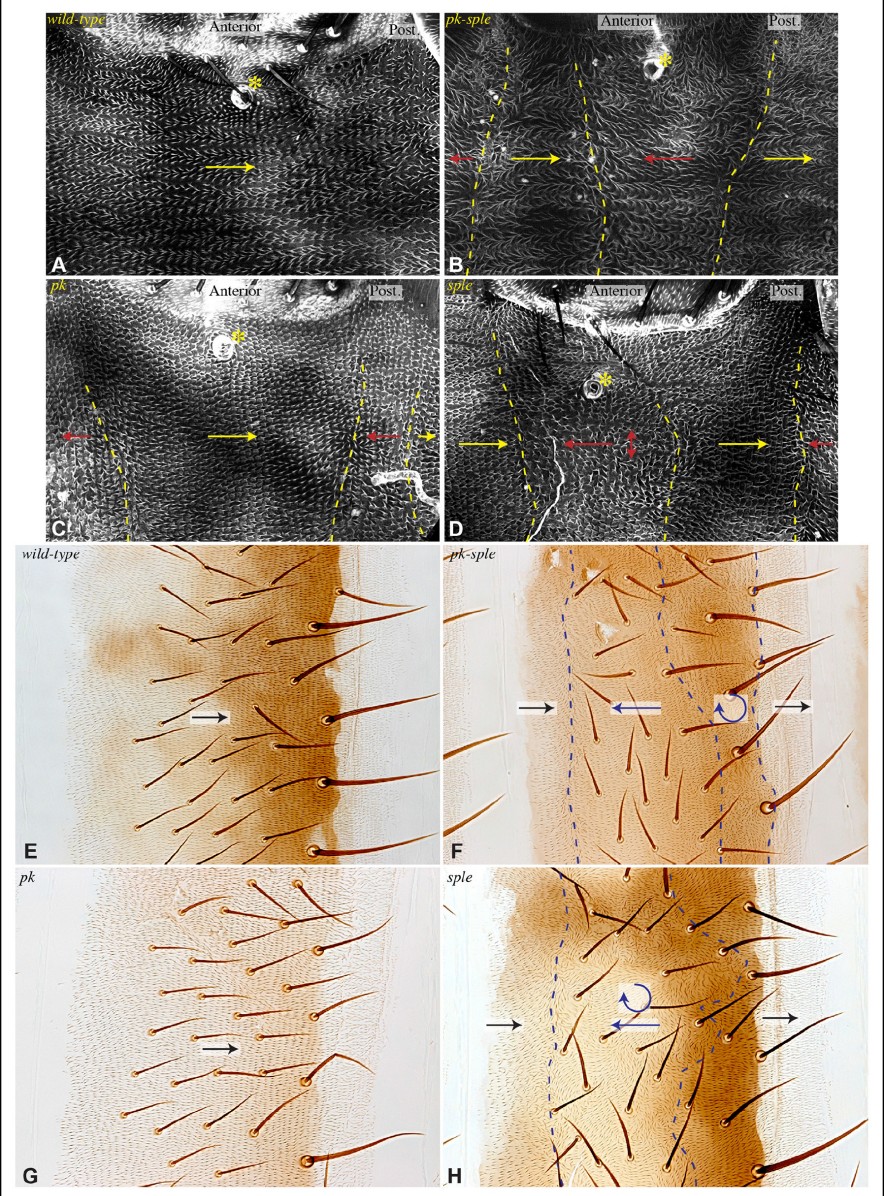

**Figure 7.** Influence of Pk and Sple on hair polarity in the abdomen. (A–D) Hair polarity in pleura revealed by F-actin (phalloidin staining) in wild type (A), and in *pk-sple*[14] (B), *pk*[30] (C) and *sple*[1] (D) mutants. Yellow asterisk indicates the position of the spiracle, which forms near the center of the anterior compartment. Yellow arrows indicate the region where hair orientation is normal, and red arrows indicate the region where hair orientation is disrupted. Dashed yellow lines mark approximate boundaries between regions with normal and abnormal polarity. (E–H) Hair polarity in tergites of wild-type (E), *pk-sple*[14] (F), *pk*[30] (G) and *sple*[1] (H) mutant animals. Black arrows indicate regions where hair orientation is normal, and blue arrows indicate the region where hair orientation is abnormal. Dashed blue line mark approximate boundaries between regions with normal and abnormal polarity.

genetic interactions and protein localization. The disruption of polarity within A compartments in *fat* or *ds* mutants was correlated with mis-localization of Dachs throughout the A compartment (mostly uniform Dachs in *fat* mutants, and randomized Dachs in *ds* mutants, **Figures 6D**, **8A,K**), and mis-localization of Sple everywhere except the most anterior region of the A compartment (**Figure 6E,H**, **8B**). Moreover, mutation of *dachs* suppressed the hair polarity phenotypes of *fat* and *ds* in A compartments (**Figures 9C,F**, **10C,G**) (**Mao et al., 2006**), and also suppressed the mis-localization of Sple (**Figures 6F,H**, **8D**). These observations suggest that *ds* and *fat* polarity phenotypes in the anterior abdomen can be accounted for by a Dachs-dependent mis-localization of Sple, as we had observed in the wing. Mutation of *dachs* alone does not disrupt polarity in A compartments

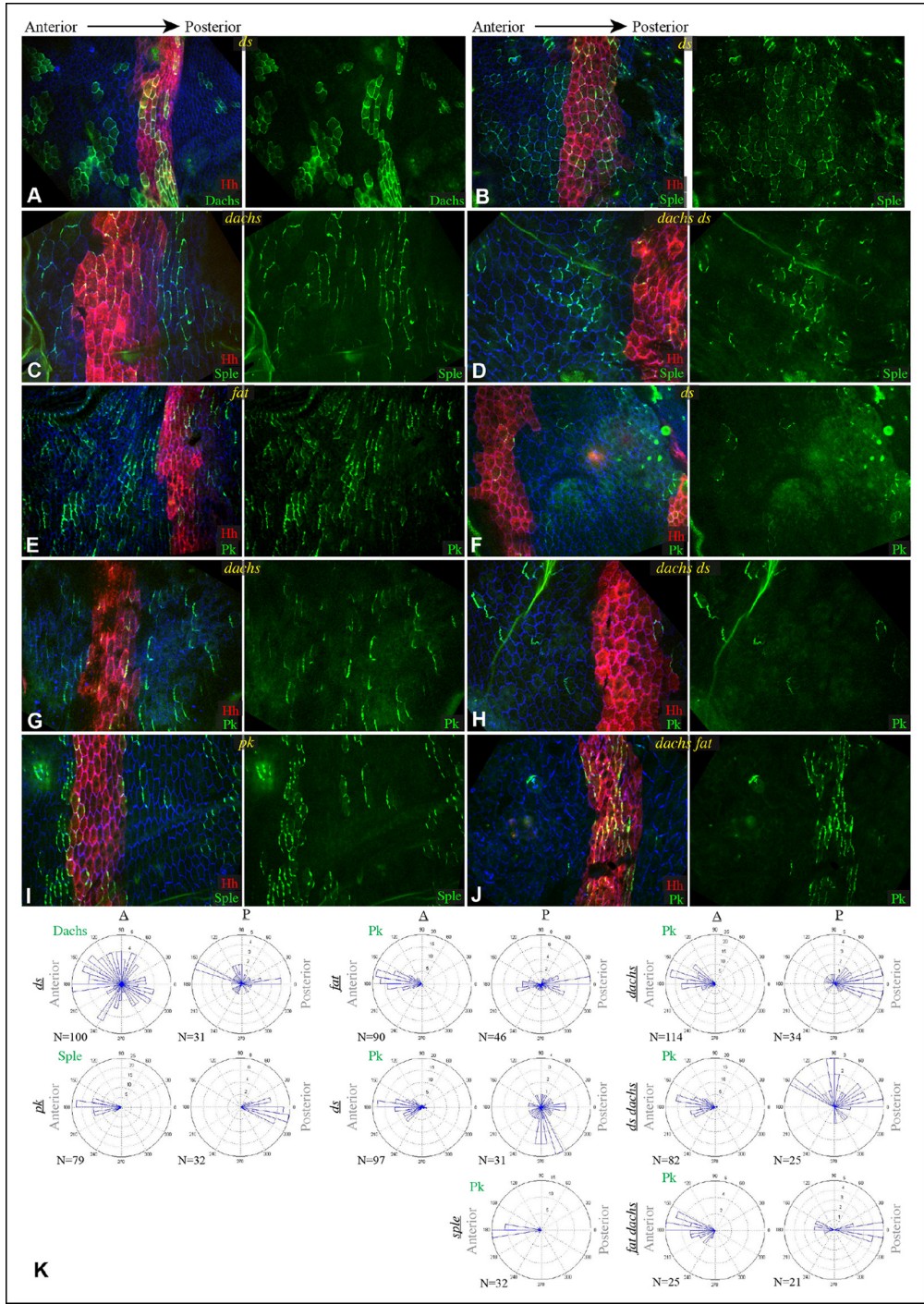

**Figure 8.** Localization of Dachs, Sple and Pk in abdominal pleura of additional genotypes. (A–J) Pleura of $ds^{36D}$/$ds^{UA071}$ (A,B,F), $dachs^{GC13}$/$dachs^{210}$ (C,G), $ds^{36D}$ $dachs^{GC13}$/$ds^{UA071}$ $dachs^{GC13}$ (D,H), $ft^8$/$ft^{G-rv}$ (E), $pk^{30}$ (I) and $ft^8$ $dachs^{GC13}$/$ft^{G-rv}$ $d^{GC13}$ (D,J) mutant pupae with clones of cells expressing of GFP:Dachs (A), GFP:Sple (B,C,D,I) and GFP:Pk (E-H,J) (green). Posterior compartments are marked by hh-Gal4 UAS-mCD8-RFP (red). (K) Rose plots depicting polarization of GFP:Dachs, GFP:Sple or GFP:Pk in pleural clones of the indicated genotypes; anterior polarization is to the left and posterior polarization is to the right. Clones were scored separately in A and P compartments

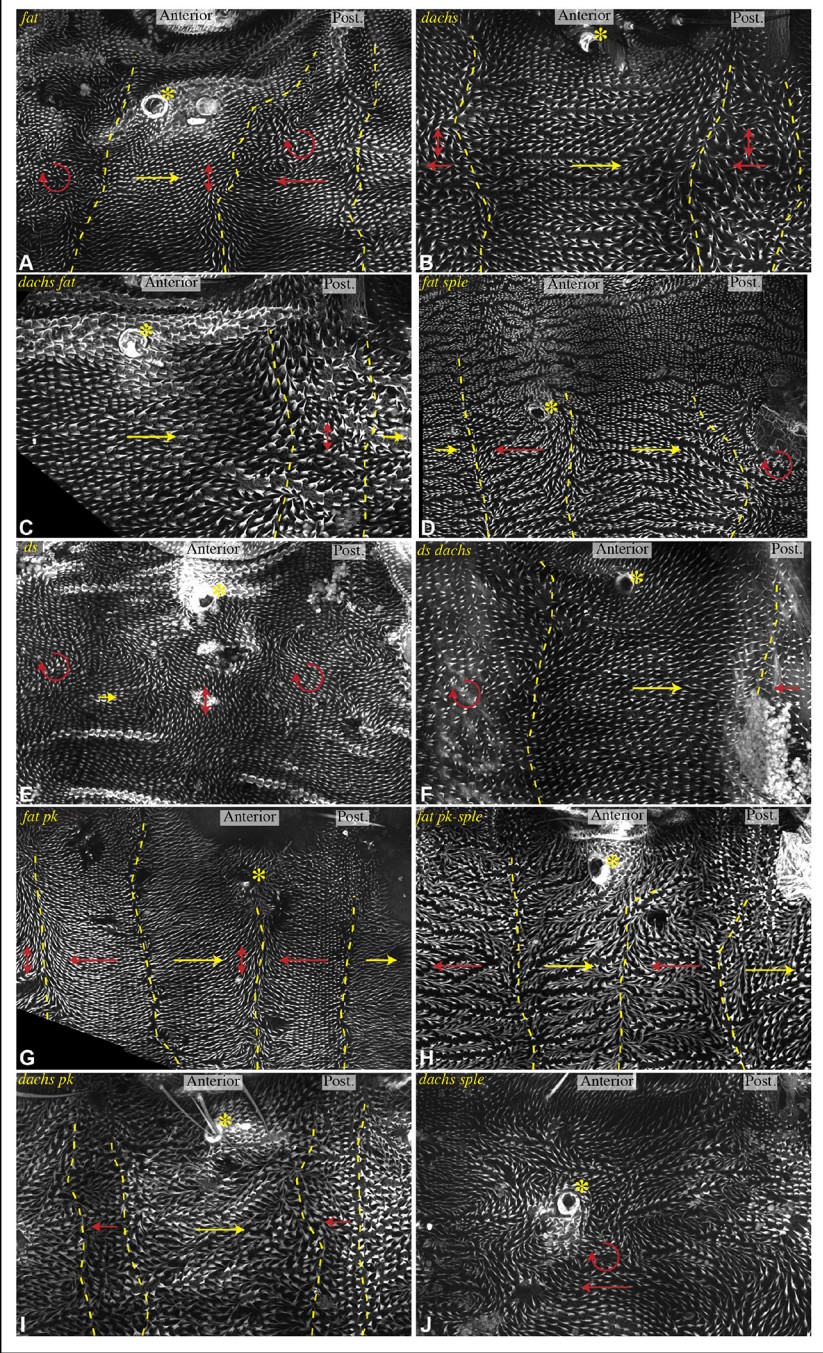

**Figure 9.** Influence of Ds-Fat PCP on hair polarity in abdominal pleura. Hair polarity in pleura revealed by F-actin (phalloidin staining) in $ft^8/ft^{G-rv}$ (**A**), $d^{GC13}/d^{210}$ (**B**), $d^{GC13}$ $ft^8/d^{GC13}$ $ft^{G-rv}$ (**C**), $ft^8$ $sple^1/ft^{G-rv}$ $sple^1$ (**D**), $ds^{36D}/ds^{UA071}$ (**E**), $d^{GC13}$ $ds^{36D}/d^{GC13}$ $ds^{UA071}$ (**F**), $ft^8$ $pk^{30}/ft^{G-rv}$ $pk^{30}$ (**G**), $ft^8$ $pk$-$sple^{14}/ft^{G-rv}$ $pk$-$sple^{14}$ (**H**), $d^{GC13}$ $pk^{30}$ (**I**) and $d^{GC13}$ $sple^1$ (**J**) mutant animals. Yellow asterisk indicates the position of the spiracle. Yellow arrows indicate the region where hair orientation is normal, and red arrows indicate the region where hair orientation is disrupted. Dashed yellow lines mark approximate boundaries between regions with normal and abnormal polarity.

(*Figures 9B*, *10B*). It could be that in the absence of Dachs and Ds, Sple is localized by the same cues that localize Pk, as Pk localization remained normal within A compartments of *ft* or *ds* mutants (*Figure 8E,F,K*).

In P compartments, Dachs is mis-localized in *fat* or *ds* mutants, and there is also a partial mis-localization of Sple (, *Figures 6*, *8*). However, mutation of *dachs* alone causes a reversal of hair

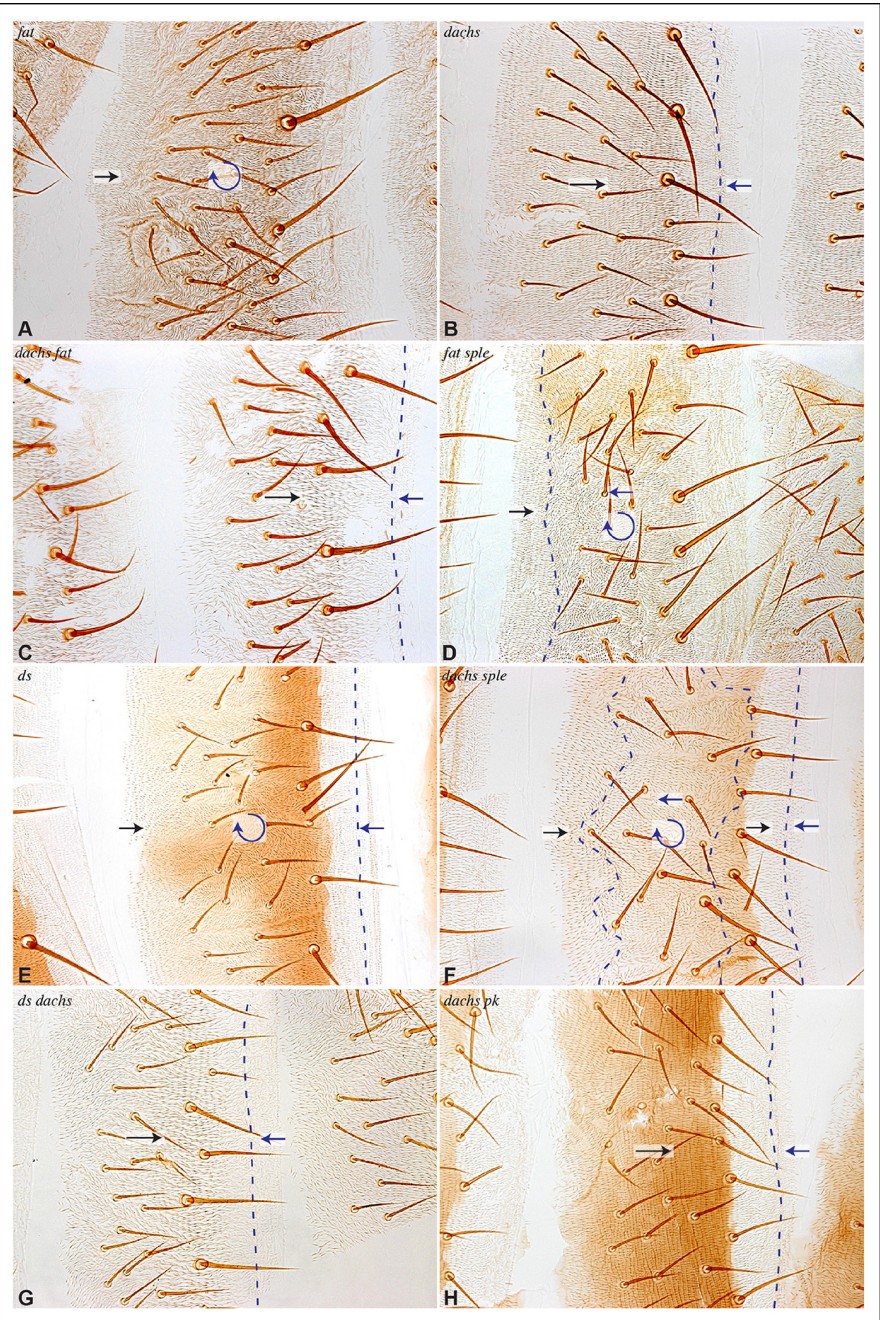

**Figure 10.** Influence of Ds-Fat PCP on hair polarity in abdominal tergites. Hair polarity in tergites of $ft^8/ft^{G-rv}$ (A), $dachs^{GC13}/dachs^{210}$ (B), $ft^8\ dachs^{GC13}/ft^{G-rv}\ dachs^{GC13}$ (C), $ft^8\ sple^1/ft^{G-rv}\ sple^1$ (D), $ds^{36D}/ds^{UA071}$ (E), $dachs^{GC13}\ sple^1/dachs^{GC13}\ sple^1$ (F), $ds^{36D}\ dachs^{GC13}/ds^{UA071}\ dachs^{GC13}$ (G), and $dachs^{GC13}\ pk^{30}/dachs^{GC13}\ pk^{30}$ (H) mutant animals. Black arrows indicate the region where hair orientation is normal, and blue arrows indicate regions where hair orientation is disrupted. Dashed blue line mark approximate boundaries between regions with normal and abnormal polarity.

polarity in P compartments (*Figures 9B*, *10B*) (*Matakatsu and Blair, 2008*). This reversal of polarity is associated with a reversal of Pk localization (*Figure 8G,K*). The P compartment of the abdomen thus differs from other regions we have examined both in that there is a strong PCP phenotype associated with mutation of *dachs*, and in the mis-localization of Pk in *dachs* mutants. One potential explanation for this could be that in the absence of Dachs, Pk localization becomes governed by

Sple, which retains its normal posterior localization in *dachs* mutants (*Figures 6G*, *8C*). In *dachs sple* mutants, Pk was partially randomized, but an overall posterior bias in localization was still observed (*Figure 6—figure supplement 1*). In *dachs sple* or *dachs pk* mutant abdomens, there is still some reversal of hair polarity in P compartments, although the region of reversal appears narrow than in *dachs* mutants (*Figure 10F,H*). Pk localization is also disturbed in P compartments of *fat* or *ds* mutants (*Figure 8E,F,K*), as well as in *fat dachs* or *ds dachs* double mutants (*Figure 8H,J,K*). Thus, while there are some similarities in control of PCP between abdominal P compartments and wings, there are also differences, hence distinct mechanisms contribute to the control PCP in each of these body regions.

## Discussion

### Sple can link PCP pathways

Our results implicate interactions between Dachs and Sple, and between Ds and Sple, as a connection point between PCP pathways coordinating polarity in multiple *Drosophila* organs. We found that both Dachs and Ds can each independently bind to Sple, but not Pk, through the unique N-terminus of Sple. Dachs and Ds also each have the ability to influence Sple localization, and we identified some places where Dachs is necessary and sufficient to localize Sple (i.e. distal wing), and others where Dachs is dispensable but Ds is required (e.g. proximal wing, eye). The influence of Ds on Sple localization is essential for some manifestations of PCP, such as ommatidial polarity in the eye. Indeed, because Fz PCP can propagate from cell to cell, we propose that localization of Sple by Ds could account for the long-range influence of Ds-Fat borders on PCP in the eye (*Sharma and McNeill, 2013*; *Strutt and Strutt, 2002*; *Yang et al., 2002*). The influence of *dachs* on PCP is generally mild, likely due to partial redundancy with Ds in localizing Sple. However, a strong influence of Dachs is revealed in the context of additional PCP mutations, as it contributes to the disturbed Sple localization and hair polarity in the wing and abdomen of *fat* and *ds* mutants, as well as the reversal of wing hair polarity in *pk* mutants.

Several mechanisms by which Pk-Sple could influence the polarization of components of the Fz-PCP system have been described. These include physically associating with Vang, and promoting clustering, endocytosis, and/or degradation of Vang or other PCP proteins (*Bastock et al., 2003*; *Cho et al., 2015*; *Jenny et al., 2005*; *Strutt et al., 2013*; *Tree et al., 2002*). They have also been identified as influencing the orientation of apical non-centrosomal microtubules that can traffic components of the Fz-PCP system, including Fz and Dsh (*Matis et al., 2014*; *Olofsson et al., 2014*). The various activities ascribed to Pk-Sple are not mutually exclusive, and it could play multiple roles. However, the only unique functions clearly attributed to Sple as opposed to Pk are its interactions with Ds and Dachs, and distinct localization. Fz and Fat localize to opposite sides of cells in wings, but to the same side in eyes (*Matis and Axelrod, 2013*) (*Figure 1A*, *Figure 5—figure supplement 1*). Because Sple expression is relatively high in eye and low in wing (*Ayukawa et al., 2014*; *Merkel et al., 2014*; *Olofsson et al., 2014*), our studies are consistent with a molecular explanation for 'rectification' of this relationship between PCP pathways in which interaction of Sple with Dachs and Ds links PCP pathways in eyes but not in wings; coupling between pathways that depends upon physical interactions with Sple and not Pk could similarly explain the opposite relationships between hair polarity and Ds and Fj gradients in A versus P abdominal compartments. Thus, the solution our results support for the controversy over the relationship between the two PCP pathways is that in some contexts they operate in sequence, with directional information passed from Ds-Fat PCP to Fz PCP via Sple, whereas in other contexts they are uncoupled.

While vertebrates have a Ds homologue that is required for PCP, Dchs1 (*Mao et al., 2011a*), Dchs1 must influence PCP in mammals through a distinct mechanism, as Pk is conserved in vertebrates, but the Sple isoform is not. Even in flies, the linkage of Dachs and Ds to Sple cannot be the sole mechanism by which Ds-Fat signaling influences PCP, as some manifestations of cell polarity controlled by Ds-Fat, e.g. oriented cell divisions, do not require Sple (*Baena-Lopez et al., 2005*; *Gubb et al., 1999*). Moreover, when mis-expressed, Ds and Fat can alter PCP even in flies lacking a functional Fz PCP system (e.g. *fz⁻ stan-* flies) (*Casal et al., 2006*). It has also been proposed that PCP in the wing is influenced by shear forces generated by contraction of the wing hinge (*Aigouy et al., 2010*), and disruption of these shear forces in *fat* or *ds* mutants might occur through a mechanism

that depends upon *dachs* but not *sple*, due either to effects of Dachs on Hippo signaling (*Cho et al., 2006*) or on cytoskeletal tension (*Bosveld et al., 2012*; *Mao et al., 2011b*). Such additional influences of Dachs might explain why *fat* wing hair polarity phenotypes are more strongly suppressed by loss of *dachs* than by loss of *sple*.

## Competition between polarizing cues

One revelation from analysis of Pk and Sple localization is that not only can PCP be oriented differently in different places or at different times (*Hogan et al., 2011*; *Sagner et al., 2012*), even at one place, cells can be subject to simultaneous, competing, polarity cues. For example, in the wing, where GFP:Sple localizes differently from GFP:Pk, cells must thus choose between competing polarity cues. Normally, they choose Pk localization cues, because Sple expression is low (*Ayukawa et al., 2014*; *Merkel et al., 2014*; *Olofsson et al., 2014*). Nonetheless, the Sple expressed in the wing is functional and able to direct PCP, as evidenced by the *dachs-* and *sple*-dependent reversals of hair polarity in *pk* mutants.

Based on observations that *pk-sple* alleles could have weaker phenotypes than isoform-specific alleles, and that over-expression of Pk or Sple could result in phenotypes reminiscent of *sple* or *pk* alleles, respectively, it was proposed that PCP requires a balance between Pk and Sple (*Gubb et al., 1999*). However, we suggest that their relationship is better described as a competition. In the wing disc, Pk expression is more abundant than Sple expression, hence Pk 'wins', and cells orient in response to cues that are unrelated to Ds-Fat PCP. When Pk is removed, then Sple can direct PCP, and hair polarity becomes governed by Ds-Fat PCP. Wild-type PCP requires Sple in some places, and Pk in others, but we know of no results that would require a balance between these two isoforms at any one place and time.

We further propose that the competition between Sple and Pk is carried out by feedback mechanisms that promote polarization. Positive feedback mechanisms, which reinforce the accumulation of co-localized proteins, together with negative feedback mechanisms, which inhibit the accumulation of oppositely localized proteins, are a staple of PCP systems, and have been widely viewed as a means of amplifying, maintaining, and propagating polarization in response to weak polarity signals. The observation that cells sometimes need to choose between competing polarity signals leads us to emphasize that feedback mechanisms could also have a distinct, fundamentally important role in PCP that has not previously been considered – they enable cells to make a discrete choice between competing polarity signals.

The observation that the relative level of Pk versus Sple influences how cells respond to competing polarity signals, with that choice then amplified by feedback, also has implications for the interpretation of GFP:Pk and GFP:Sple localization profiles. We take the localization of these proteins as indicators of the polarity signals that cells 'see' when that isoform predominates. This is not necessarily the same as their localization under endogenous expression conditions. For example, endogenous Pk localization might normally match Sple in the eye even in front of the furrow, because it is recruited to equatorial sides of cells by interactions with Sple and Vang.

## Influence of Ds-Fat signaling on PCP in the wing

Analysis of wing hair polarity played a central role in development of the hypothesis that Ds-Fat functions as a 'global' PCP module and Fz as a 'core' PCP module, with polarity guided by the vectors of Fj and Ds expression (*Ma et al., 2003*). However, since Ds-Fat signaling modulates Sple, but not Pk, localization, and Pk, but not Sple, is normally important for wing hair polarity, we infer that Ds-Fat PCP does not normally play a significant role in directing wing hair polarity. Instead, we propose, as also suggested by (*Blair, 2014*), that the hair polarity phenotypes of *ds* or *fat* mutants are better understood as a de facto gain-of-function phenotype, resulting from inappropriate accumulation of Dachs on cell membranes, which then leads to inappropriate localization of Sple, and abnormal polarity. This would also explain how Ds-Fat signaling, stripped of polarizing information, could nonetheless rescue PCP phenotypes: for example, how uniform Ds expression can rescue hair polarity in *ds fj* mutants (*Matakatsu and Blair, 2004*; *Simon, 2004*), and how expression of the intracellular domain of Fat can rescue hair polarity in *fat* mutants (*Matakatsu and Blair, 2006*), as these manipulations suppress the membrane accumulation of Dachs that would otherwise occur in mutant animals.

More recently, it has been proposed that Ds-Fat PCP provides directional information to orient Fz PCP in the wing by aligning and polarizing apical non-centrosomal microtubules that can traffic Fz and Dsh (*Harumoto et al., 2010*; *Matis et al., 2014*; *Olofsson et al., 2014*). While disorganization of these microtubules is observed in *fat* or *ds* mutants, we suggest that the inference that Ds-Fat thus orients PCP in wing via these microtubules is incorrect. There is evidence both in imaginal discs and in axons that Pk-Sple can orient microtubules (*Ehaideb et al., 2014*; *Olofsson et al., 2014*). Sple is mis-localized in *fat* or *ds* mutant wing discs. Thus, we propose that the effects of *ds* and *fat* mutants on microtubules within the wing are likely a consequence of abnormal Sple localization, which disrupts microtubule orientation, but need not be interpreted as evidence for a normal role of Ds-Fat signaling in orienting microtubules or Fz PCP in the wing.

### Instruction of PCP in the abdomen

The A compartment of the abdomen can be compared to the eye (Sple-dependent), and the P compartment of the abdomen can be compared to the wing (Pk-dependent). However, we also observed striking differences in the apparent influence of Ds-Fat pathway mutations on PCP between the abdomen and other organs. One possible contribution to these differences is the metamerism of the abdomen, and the local propagation of PCP. For example, the disturbance of Pk localization in P compartments even in *fat dachs*, *ds dachs* or *dachs sple* mutants would not be predicted based on the lack of detected physical interaction of Ds or Dachs with Pk, and lack of influence of these mutations on Pk in the wing. However, the P compartment is adjacent to two A compartments, and disturbances of polarity in these neighboring A compartments could potentially spread into the P compartment. The relatively normal polarity within A compartments of *fat dachs* or *ds dachs* mutants would also not be predicted if Sple is normally establishing polarity in response to the polarization of Ds and Dachs, and this result is not easily explained by hypothesizing propagation of normal polarity from neighboring compartments, since polarity is partially reversed in P compartments of these genotypes. Instead, we propose that in the absence of the primary PCP signal (in this case provided by the Ds-Fat pathway), cells use information from secondary signals, which appear to provide polarizing information in the A compartment that parallels that provided by the Ds-Fat pathway.

## Materials and methods

### *Drosophila* stocks and crosses

For investigation of Dachs, Sple or Pk localization we used *act> y+>EGFP-dachs /TM6b*, *act> y +>EGFP-sple /TM6b*, or *act> y+>EGFP-pk /TM6b* (*Bosveld et al., 2012*; *Strutt et al., 2013*). Clones with posterior compartments marked were made by crossing to *y w hs-FLP[122]; If/CyOGFP; hh-Gal4 UAS-mCD8-RFP /TM6b*. Mutant backgrounds examined in clones were $ft^{G-rv}/ ft^8$, $d^{210}/ d^{GC13}$, $ds^{UA071}/ ds^{36D}$, $d^{GC13} ft^{G-rv}/ d^{GC13} ft^8$, $d^{GC13} ds^{UA071}/d^{GC13} ds^{36D}$, $sple^1/ sple^1$, $d^{GC13} pk^{30}/ d^{GC13} pk^{30}$, $vang^{stbm6}$. Flip-out clones were induced by heat shock at 33°C either 24 hr (wing disc, eye disc) or 12 hr (abdomen) before dissection. Additional mutant backgrounds were $pk^{30}/ pk^{30}$, and *pk-sple*$^{14}$/ *pk-sple*$^{14}$. UAS-RNAi-fat (vdrc 9396) or UAS-RNAi-dachs (vdrc 12555) was expressed using *nub-Gal4* or *C765-Gal4* along with *UAS-dcr2*.

### Immunostaining and fluorescent imaging

Tissues were fixed in 4% paraformaldehyde in PBS followed by permeabilization in PBS with 1% BSA and 0.1% Triton X-100. Primary antibodies used include rat anti-E-cad (1:200, DSHB), anti-Pros (1:200, DSHB), anti-Elav (1:200, DSHB) and mouse anti-Wg (1:400, DSHB). Secondary antibodies used were labeled with DyLight405, Cy3 or Alexa647 (1:100, Jackson Immuno Research, West Grove, PA); GFP and RFP were detected by autofluorescence. Alexa488-phalloidin (1:10, Life Technologies) was used to stain hairs in pleura. Protein localization in pleura was determined at ~48 hr after pupal formation. Images were captured on a Leica TCS-SP5 confocal microscope or PerkinElmer Velocity spinning disc confocal microscope.

## Plasmid constructs and primers

Sple was amplified by PCR from pUAST-pk^sple (*Gubb et al., 1999*) using forward primer (5'- CTCGA-ACCACGGCGGCCGCCAACATGAGCAGCCTGTCAACCGGTGGAG -3') and reverse primer (5'- GT-GGTTCGAGGGTACCCGAGATGATGCAGTTCTTGTCCTTG -3') and cloned using *Not*I and *Kpn*I sites into pUAST-TM-EGFP:3XFlag (*Mao et al., 2009*) after removal of TM-EGFP by *Not*I/*Kpn*I digestion to generate pUAST-sple:3XFLAG. Sple(N) (first 349 amino acids) was amplified by PCR from pUAST-pk^sple using forward primer (5'- ACTCTGAATAGGGAATTGGGAATTCCAACATGAGCAGCC-TGTCAACCGGTG -3') and reverse primer (5'- GTAGTCGCCTCGAGCCGCGGCCAGCTCATTTGAC-TCCTGCTGGGCG -3') and inserted using InFusion cloning kit into pUAST-app:3XFlag2XStop (gift of B. Staley) after removal of TM-EGFP by *Eco*RI/*Not*I digestion, to generate pUAST-sple:3XFLAG. Pk was amplified by PCR from pUAST-pk^pk (*Gubb et al., 1999*) using forward primer (5'- ACTCTGAA-TAGGGAATTGGGAATTCCAACATGGATACCCCAAATCAAATGC -3') and reverse primer (5'- GTA-GTCGCCTCGAGCCGCGGCCAGCGAGATGATGCAGTTCTTGTCC -3') and inserted using InFusion cloning kit into pUAST-app:3XFlag2XStop after by EcoRI / NotI digestion, to generate pUAST-pk:3XFLAG2XStop.

## Co-immunoprecipitation and Western blotting

Tagged isoforms of Dachs, Ds-ICD, Sple, Sple-N, Pk and GFP (control) were expressed in S2 cells by transient transfection using Effectene of plasmids pUAST-attB-d:V5,His (*Mao et al., 2006*), pUAST-TM-DS-ICD:FLAG,V5,His (*Mao et al., 2009*), pUAST-attB-sple:3xFLAG, pUAST-attB-sple(N):3xFLAG-2xStop, pUAST-attB-pk(N):3xFLAG-2xStop, pAc-3XFLAG:GFP (*Oh et al., 2009*), pAc-GFP:V5 (*Feng and Irvine, 2009*). Cells were harvested 48 hr after transfection and lysed in RIPA (50 mM Tris-HCl, pH 8.0; 150 mM NaCl; 1% NP-40; 0.5% Sodium deoxycholate; 0.1% SDS; 1 mM EDTA; 1 mM DTT and 10% glycerol, supplemented with protease inhibitor cocktail (Roche) and phosphatase inhibitor cocktail (CalBiochem)). Cell lysates were precleared by incubation in 10 μl Protein-A beads for 2 hr at 4°C and later incubated with 10 μl anti-V5 beads at 4°C overnight for co-immunoprecipitation. Anti-V5 beads were washed four times with RIPA, boiled in Laemmli sample buffer at 100°C for 5 min, and run on SDS-PAGE gels. Western blotting was performed using rabbit anti-V5 (1:5000, Bethyl Labs) and mouse anti-Flag (1:3000 Sigma), and fluorescentconjugated secondary antibodies (Odyssey).

## Quantification of polarity

Polarity vectors were determined manually by estimating the vector of Sple, Pk, or Dachs polarization within a cell, and comparing it to a reference vector. The images analyzed were projections through several confocal sections. Wing discs were examined at mid- to late third instar, eye discs were examined at late third instar, and abdomens were examined at 48 h after puparium formation. Vectors of polarization were determined within scoreable cells. Scoreable cells included single cell clones, and cells within small clones, or irregular portions of larger clones, with unlabeled cells on two or three sides, and for which any GFP observed could be clearly assigned to a single cell. Vectors of polarization were drawn from the center of the cell to the strongest visible accumulation of GFP-tagged proteins. In cases where a broad region of similar intensity was observed, vectors were drawn toward the center of membrane GFP accumulation. Counterstaining with E-cadherin was employed to ensure that all junctions of scored cells were visible in the images. In wing discs, polarity was determined separately in distal, proximal and A-P boundary regions, using Wg and Hh expression as references. The proximal region was defined as cells within 5 cells of the fold at the edge of the wing pouch. The reference vector for the proximal region was a line perpendicular to the tangent of the proximal Wg ring at the point closest to the cell being scored. The A-P boundary region was defined as cells within 5 cell diameters anterior to the edge of Hh expression. The reference vector for the A-P boundary region was a line drawn parallel to the A-P boundary. The distal region was defined as cells within the wing pouch, excluding the proximal and A-P boundary regions, and also limited to the central three quarters of the D-V Wg stripe (*Figure 2I*). Cells overlapping the D-V boundary were excluded from analysis. The reference vector for the distal region was a line perpendicular the D-V boundary Wg stripe. In eye discs, the reference vector was a line parallel to the tangent of the morphogenetic furrow (the poles of the eye disc, where the morphogenetic furrow is not perpendicular to the equator, were not analyzed). In the abdomen, the reference vector

was a line perpendicular to the A-P compartment boundaries. Cells in different regions were scored separately as indicated in the figure legends, based on observed regional differences in polarity in certain genotypes. In all tissues, the angle between the vector of polarization and the reference vector was calculated using ImageJ, and rose plots summarizing the distribution of angles were generated using Matlab.

## Acknowledgements

We thank the Developmental Studies Hybridoma Bank, the Bloomington stock center, and D Strutt, S Collier, D Gubb, R Holmgren, B Staley for plasmids, antibodies, and *Drosophila* stocks, Y Feng for *fat tub-Gal4 UAS-wts* wings, Idse Heemskerk and Sebastian Streichan for help with image analysis, and Gary Struhl and anonymous reviewers for comments on the manuscript. This research was supported by NIH grant R01 GM078620 and the Howard Hughes Medical Institute.

## Additional information

### Funding

| Funder | Grant reference number | Author |
|---|---|---|
| Howard Hughes Medical Institute | | Kenneth D Irvine |
| National Institute of General Medical Sciences | R01 GM078620 | Kenneth D Irvine |

The funders had no role in study design, data collection and interpretation, or the decision to submit the work for publication.

### Author contributions

AAA, Conception and design, Acquisition of data, Analysis and interpretation of data, Drafting or revising the article; KDI, Conception and design, Analysis and interpretation of data, Drafting or revising the article

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
