## [Decision Letter]

Thank you for submitting your work entitled ‘Coordination of planar cell polarity pathways through Spiny legs’ for peer review at *eLife*. Your submission has been favorably evaluated by Fiona Watt (Senior Editor), Jeremy Nathans (Reviewing Editor), and three reviewers.

The reviewers have discussed the reviews with one another and the Reviewing editor has drafted this decision to help you prepare a revised submission.

All of the reviewers were impressed with the importance and novelty of your work. We are including the three reviews at the end of this letter, as there are multiple specific comments in them that will not be repeated in the summary here. These comments focus on a set of technical issues including the strains used, how the tissue was scored, and the consistency of phenotypes. As is the style at *eLife*, there was a discussion among the reviewers regarding the various points raised in the primary reviewers. At the risk of burdening you with excessive feedback, we have appended a somewhat truncated version of that discussion after the reviews.

While this is not the typical *eLife* style, we think the high quality of the feedback justifies giving it to you in largely unedited form.

*Reviewer #1:* The results presented here make a valuable addition to the field. Some of it repeats findings from Ayukawa et al., and I find myself wishing there was a bit more discussion of the similarities and contrasts in the Discussion section, rather than limiting these to the Results. I also thought that some of the quotations were a bit incomplete (see below). However, there is clearly enough new data here to be worth publishing, with a few revisions.

The weak co-IP that Ayukawa find to Pk is probably worth mentioning in the Discussion, even if the authors do not see it in their conditions, as it could be part of a Sple-independent mechanism.

Another result of Ayukawa's this is worth discussing is the improvement of *sple* overexpression PCP defects in a *ds* or *fat* knockdown. Shouldn't the plentiful and now-randomized Dachs still be misdirecting Sple? Why does this weaken the phenotype?

The hypothesis that inappropriate accumulation of Dachs, and thus inappropriate coupling to the core pathway, was part of the fat mutant PCP phenotype, was I think first proposed in Blair's 2014 Dispatch. I wonder if it quite works, however, to explain the rescuing effects of unpatterned Fat-Ds in the wing, as the current manuscript proposes, in light of Matakatsu and Blair 2012. They showed Fat constructs that rescued Hippo signaling, and thus presumably rescued heightened Dachs accumulation, but that did a rather poor job rescuing PCP. They also had constructs that did not rescue Hippo, and these did a much better job improving abdominal PCP.

Specific notes:

Results:

The statements do not make it clear that Ayukawa did see co-IP between Ds and Sple-N. Ayukawa also went a bit further, and showed that binding depended on the very N terminus, and was to a middle region of the Ds ICD.

Reduced junction accumulation of Ds in Fat mutants – the authors should also reference Strutt and Strutt 2002, and Ma et al. 2003.

The partial rescue of *fat* mutant abdominal PCP defects by loss of *dachs* has already been published in Mao et al. 2006, admittedly not in the detail presented here. Similarly, improvement of *ds* mutant wing PCP defects by loss of *dachs* was in Brittle et al. 2012.

Since *dachs* loss will affect both Sple and the Hippo pathway, can the authors show that the *dachs fat* effect on wing PCP is different than the rescue of *fat* PCP achieved by increasing Hippo pathway activity in *fat* mutants, as claimed in Feng and Irvine, 2007?

"Thus, behind the morphogenetic furrows…" Strictly speaking, the authors have shown that protein polarization depends on Ds, Ft or Sple, but not that these direct it. To do that, they would have to change their levels or, better, repolarize them. They have also not shown that these effects depend on the Ds ICD.

Methods and figures:

The Methods say far too little about how polarity was measured in clones with tagged proteins, only that it was **‘**manual’ and used ImageJ. What was being measured, and how did that translate into the rose plots? Was there an assessment of the intensity of polarization in a clone, or just a direction for each clone? How was that determined and quantified? Did this assessment only use cell faces at clone boundaries, or internal cell faces? Did it adjust for the shapes of the cells and clones, and thus the amount of membrane in each quadrant? Since proximal and distal cell faces can be at slightly different focal planes due the folding of the pouch, was more than one confocal slice used for the image?

And two anatomical questions for the wing disc results: By ‘distal’ and ‘proximal’, do the authors mean the distal tip, or the prospective wing margin? Second, did they have any objective landmarks for separating proximal from distal portions of the wing pouch? The fate map can be a bit tricky if they are only using the fold, as the tissue close to the fold can change depending on the stage of eversion, especially near the AP boundary. One could always try using the identifiable sensory organ precursors along the third vein. It would be helpful to know where this actually maps to in an adult wing.

*Reviewer #2:* The manuscript by Ambeganokar and Irvine addressed how two different pathways that regulate planar cell polarity in *Drosophila* are coordinated. Recent work by a number of groups has implicated *pk/sple* as being a key in coupling these two pathways and this current manuscript adds to this emerging story. There are several valuable insights that come from this manuscript. A key finding is that Sple can interact physically with both Dachs and Ds. The authors provide a new way to view several issues. They describe a series of genetic experiments that show that Dachs mediates the localization of Sple in wing cells and that in regions where Ds is highly expressed it can independently direct the localization of Sple. They provide evidence for an alternative model for the *ds/ft* wing hair phenotype where this is viewed as a consequence of inappropriate *dachs* localization which results in altered accumulation of Sple which can in turn interfere with Pk function. The paper also provides an explanation for several puzzling results from the literature: the ‘rectification’ of the relationship between *fz* and *ds/ft* and for the ability of the uniform Ds expression to rescue the hair polarity of *ds fj* mutants. They also describe some interesting new observations that have been missed by others (e.g. that Pk localization is not proximal in all regions of the wing) even if the significance of the observation is unclear.

Specific comments:

In the subsection ‘Control of Sple polarity in eye discs by Ds-Fat PCP’, Figure 5. It is not clear to me how the authors conclude that the Sple in panel A (left side) is in R4 and in panel D (right side) R3.

Figure 5—figure supplement 2. I did not find this figure to be convincing (particularly B).

Figure 7, Figure 7. The location of the spiracle allows the authors to determine the middle of the A compartment in the pleura but it is not clear to me how they determine where the AP boundary is. Are they counting cells?

In the subsection ‘Competition between polarizing cues’, the authors argue that the previous *pk/sple* balance model is not supported by the data and that a competition model is preferable. I agree with the conclusion that we should now think about the *pk/sple* interaction as a competition but I see that as providing a mechanism for the balance model.

*Reviewer #3:* In this manuscript, Ambegaonkar et al. address the function of the Core PCP Pk/Sple locus in regulating the coupling between the Fat and Core PCP systems.

Earlier work had suggested that Pk and Sple isoforms influence the coupling of the Core PCP and Fat PCP systems. While Sple enforces coupling such that Vang and Ds/Dachs are on the same side of the cell, Pk allows coupling in the opposite orientation or even complete uncoupling of Core and Fat polarity, allowing Core PCP to respond to other orienting cues. Biochemical experiments had suggested that Sple interacts directly with Dachs. These questions had been most thoroughly explored in the wing, but less was known about coupling of Core and Fat PCP in other tissues like the eye and anterior abdomen where Sple function is more important.

The current manuscript from Ambegaonkar et al. now extends these studies to the eye and abdomen. It also reveals a second interaction between Sple and Dachsous, and shows that Sple is influenced by Dachs and Ds in different places. The authors map the localization of Pk and Sple in the wing disc, eye disc and pupal abdomen by generating clones of fluorescently tagged Pk and Sple isoforms that are over-expressed under the control of the actin promoter. They examine how perturbations of Fat system components influence the localization of PkGFP and Sple:GFP, and correlate these with polarity phenotypes. Their data show that PkGFP and Sple:GFP clones have opposite polarity in wing discs and posterior abdomen (where Pk normally dominates), but the same polarity in eye discs posterior to the furrow and the anterior abdomen (where Sple normally dominates). This is an intriguing correlation, but the authors data do not explore or clarify why Pk and Sple localize together in some tissues and oppositely in others. Their data generally confirm the idea that the Fat system orients Sple in the eye disc and wing disc, but the data in the abdomen are very confusing and no clear model emerges for the orientation Sple or Pk here.

In general, I think some of the confusion may stem from the clones used to map Sple and Pk polarity, and from the unusual organization of the abdomen – here Sple and Pk-dependent regions are directly opposed to one another. There are several experiments the authors could do to resolve this confusion that should be possible within the time frame of a revision.

Pk and Sple polarity:

Throughout the, the authors map Pk and Sple polarity using flip out clones of GFP tagged Pk or Sple that are over-expressed under the control of the actin promoter. They use these constructs to deduce the localization of these proteins even in tissues where the particular isoform in question is expressed at much lower levels than the other – for example they use GFP-Sple clones to make conclusions about Sple localization in the wing disc even though endogenous levels are hardly detectable – in any case much lower than Pk. But given the feedback mechanisms that lead to polarized formation of Core PCP domains, and the fact that these proteins are overexpressed, this may not reflect the endogenous localization. What I mean is that the very small amount of Sple that is normally present in the wing disc might ordinarily be incorporated into proximal Core complexes that are oriented by the much larger amounts of Pk and the cues read by Pk. The distal localization of Sple in the wing disc (as visualized by actin promoter-driven over-expression of Sple:GFP) could result from Core PCP reversal caused by Sple over-expression. They conclude that wing cells normally have Pk on one side and Sple on the other, but this may not be true at endogenous levels.

To rule this out, the authors should stain for other Core PCP components to be sure that these Sple:GFP clones do not disturb the polarity field, and look at adult flies to make sure these clones don't cause polarity defects.

If Sple:GFP really localizes proximally in the wing without otherwise disturbing the polarity of Core PCP domains, it would suggest that Sple (unlike Pk) doesn't need other Core PCP proteins to localize to the membrane – not unreasonable if Sple interacts with Ds and Dachs. This would be easy to show by visualizing Sple in the background of *fz* or *vang* mutants – if it is localized by Ds and Dachs, it should still be found on the proximal side of cells in the wing.

Studies in the abdomen:

No clear conclusions emerge from the studies on the abdomen. It is puzzling that *dachs* mutation suppresses mislocalization of Sple caused by loss of fat or ds in the anterior abdomen. If Sple is normally oriented by *ds* or *dachs*, where does the information now come from? Furthermore, in the Discussion, the authors say that Fat-Ds signaling modulates Sple but not Pk localization. However their data in the posterior compartment of the abdomen seem to suggest that Pk can also read cues provided by the Fat PCP system (but somehow with the opposite polarity as Sple). There are no identified physical interations between Pk and Fat components and why it should be oriented in this way by Fat PCP in the posterior abdomen but not the anterior abdomen or eye is puzzling. Furthermore, this must not always be true because sometimes Pk is oriented completely orthogonally to Fat PCP (in the pupal wing and in the eye disc before the furrow passes.

The abdomen is complex because two regions with different requirements for Pk and Sple lie directly adjacent to one another. Domineering non-autonomous effects of Core PCP likely propagate over some distance through these regions and rescue (or disrupt) Core PCP domain polarity and hair polarity in other regions. This could mask where specific Fat and Core components are actually needed. This could explain why removing Dachs suppresses the polarity defects caused by Fat or Ds mutation in the anterior abdomen – once all signals from Fat PCP are gone, then correct Core polarity, propagating in to the anterior compartment from the posterior compartment, could correctly orient Sple and hair polarity. A similar mechanism might explain why Pk localization in the posterior is disturbed by loss of Fat system components – i.e. Sple-dependent alterations of Core polarity in the anterior are propagating via Core PCP interactions into the posterior.

To clarify this, the authors should target RNAi of PCP components to specific subregions of the abdomen to find out where their effects are direct and where they occur as a consequence of domineering non-autonomy.

More specific comments:

1) The authors cite both Hogan et al. and Merkel et al. as saying that there is an early phase of polarity establishment in the wing where Sple is reading a Fat/Ds-dependent signal. But the Merkel et al. paper actually says the opposite – this paper shows that loss of Sple has no consequences for Core PCP orientation in the wing until very late after hair formation when the ridge pattern is generated. The idea that Sple has an early role was based on over-expression experiments.

2) The cartoon in Figure 1 suggests a proximal distal orientation of the Fat PCP system in the pupal wing, but this is not the case. Four-jointed is expressed around the whole margin and Ds is elevated at the AP boundary as well as the hinge – the polarity pattern of the domains themselves is first radial and then AP oriented in pupal wings. Since the data they show in Figure 1 pertain to the wing disc not the pupal wing, maybe it would be better to show a cartoon of the Fat PCP pattern in the wing disc where it is quite different.

3) In Figure 4, the authors should quantify the hair polarity phenotypes indicating suppression of the *fat* mutant by *sple* – a single example is not enough.

4) In Figure 6, the authors should indicate the anterior posterior axis on the images. Also, it is important to show wild type polarity rose diagrams with the same breakdown into Af, Am and Ab or A, A* and P as they show for the mutants.

5) In the Discussion, I think the authors are making what seems to be a semantic distinction between a ‘balance’ of Pk and Sple (as suggested originally by David Gubb) and ‘competition’ between Pk and Sple supported by their results. The conclusions are really the same – that Pk and Sple antagonize each other and that phenotypic disturbances when one isoform is lost are caused by inappropriate (or unbalanced) activity of the other.

Additional discussion comments:

I would have liked to see more discussion of the similarities and differences between the current manuscript and Ayukawa. Of course it is possible that they do not have any substantial insights into the differences – if so additional comments might not be valuable.

I also agree the manuscript would be substantially improved by a more detailed discussion of how they measured polarity. A supplemental figure showing an example would be good.

The Strutt et al. paper that describes the Sple:GFP and PkGFP constructs shows that they do disrupt polarity when expressed uniformly in the inappropriate tissues. The Sple:GFP construct causes a *pk*-like wing hair orientation pattern, and the PkGFP construct disturbs omatidial polarity in the eye. I would be surprised if the situation were different in the clones used in this paper, and I think it is essential to check the polarity around these clones. In any case, I think it will be hard to argue that Sple would localize proximally when expressed at endogenous levels in a tissue that is dominated by Pk. The reverse isn't as much of a problem – even though PkGFP is over-expressed in clones, it still generally colocalizes with Sple in the eye and anterior abdomen where Sple dominates. This makes it a very interesting question why PkGFP expression causes polarity defects in the eye as reported by Strutt et al. It would seem to suggest that some function of Pk itself, rather than reversal of Core PCP domains, is responsible for Pk over-expression defects in the eye – what do you guys think?

I do also think it is important to target knockdown of different Fat pathway components to specific abdominal regions. At the moment, the results of these experiments are confusing and inconsistent with the results from other tissues – no clear model can be drawn from them. Of course sometimes that's just the way life is, but non-autonomous effects would be a trivial explanation for these discrepancies and I think they should be ruled out so as not to confuse the field needlessly.

I agree that they need to test the effects of the flip-outs.

As for the confusing nature of the abdomen, they do provide some possible hypotheses in the Results. In the case of the anterior, Reviewer 3 was asking what supplied the information for normal Sple polarity in the *ds dachs* double. The manuscript opines that it could be whatever localizes Pk, presumable meaning some part of the core system, since Pk is pretty normal. Or I suppose one could invoke Pk-Sple binding. They hypothesized both to explain proximal Sple in *ds dachs* double mutant wings, so it is pretty much the same idea here.

The sensitivity of Pk to *ds* or *dachs* removal in the posterior is bit more confusing, as I think they admit. What they suggest is that, in absence of Ds cues, now Pk becomes sensitive to Sple and follows mislocalized Sple. It could be explained a bit better, as the molecular interactions of Dachs and Ds are with Sple; I guess this unbound Sple is now more attractive to Pk? Or we could invoke the weak Pk-Dachs binding of Ayukawa, but it would have to be binding that mediate repulsion or destruction, to get anterior Pk and posterior Dachs.

At any rate, they did not look at Pk localization in a *ds sple* or *dachs sple* double, if they are really trying to support this idea. And if we're being complete, they must know what happens to Sple in a *ds dachs* double in the posterior. If Sple gets strongly reversed like it does in *ds dachs* wing, I'm not sure it can account for the Pk mislocalization they note in a *ds dachs* double.

On the other hand, I would bet in a *ds* or *fat* mutant wing that, if you waited long enough, you would get misorientation of Pk polarization, just like you do of the other core polarity proteins in a 28–32 hr AP wing. What stage of abdomen were they looking at, and how long before hair formation?

I have to agree that about the need for examining the polarity around the clones. I think that rises to the level of essential. It is hard to imagine that they did not at least look at adult wings for polarity phenotypes.

---

## [Author Response]

Reviewer #1:

*The results presented here make a valuable addition to the field. Some of it repeats findings from Ayukawa et al., and I find myself wishing there was a bit more discussion of the similarities and contrasts in the Discussion section, rather than limiting these to the Results. I also thought that some of the quotations were a bit incomplete (see below). However, there is clearly enough new data here to be worth publishing, with a few revisions. The weak co-IP that Ayukawa find to Pk is probably worth mentioning in the Discussion, even if the authors do not see it in their conditions, as it could be part of a Sple-independent mechanism.* As we were unable to detect this interaction (and think it is most likely an artefact of non-specific association rather than a real interaction), we don't have anything further to say beyond noting, as we do, that this is a difference between our results and those of Ayukawa et al.

*Another result of Ayukawa's this is worth discussing is the improvement of* sple *overexpression PCP defects in a* ds *or* fat *knockdown. Shouldn't the plentiful and now-randomized Dachs still be misdirecting Sple? Why does this weaken the phenotype?*

Sple over-expression results in a reversal of polarity. According to our results and model, this is because Sple is localized by Dachs. In otherwise wild-type wings, Dachs directs this over-expressed Sple to the distal sides of cells, reversing polarity. Upon fat or ds knockdown, Dachs is now uniform. Hence, co-localization of Sple with Dachs no longer specifies a reversal of polarity, and this phenotype is suppressed. Whether this should lead back to normal polarity (Sple gives no net directional cue, but Pk does, hence cells follow the Pk cue) or randomized polarity (Sple is on all sides of cells, and cells follow the Sple cue because its over-expressed) depends upon how cells resolve this situation – we would have predicted the latter, but it’s possible that it’s the former. The Ayukawa et al. paper only actually shows the polarity in a very small region in the center of the wing (boxed region in their Figure 3) so it’s actually not clear what the overall result is here, and hence it’s difficult for us to discuss.

*The hypothesis that inappropriate accumulation of Dachs, and thus inappropriate coupling to the core pathway, was part of the* fat *mutant PCP phenotype, was I think first proposed in Blair's 2014 Dispatch.*

Thank you for pointing this out, we now cite this review in the Discussion of the revised manuscript.

*I wonder if it quite works, however, to explain the rescuing effects of unpatterned Fat-Ds in the wing, as the current manuscript proposes, in light of Matakatsu and Blair 2012. They showed Fat constructs that rescued Hippo signaling, and thus presumably rescued heightened Dachs accumulation, but that did a rather poor job rescuing PCP. They also had constructs that did not rescue Hippo, and these did a much better job improving abdominal PCP.*

We have also identified Fat constructs that preferentially rescue Hippo or PCP signaling (Pan et al. 2013). In our case, we found that this correlated with whether these regions primarily affected polarization of Dachs, or levels of Dachs, consistent with our earlier studies arguing that Hippo is sensitive to the total amount of Dachs on membranes, whereas PCP is sensitive to the direction of Dachs polarization. Matakatsu and Blair 2012 did not examine Dachs localization, without knowing this, it is not possible to make firm conclusions regarding the role of Dachs in the phenotypes of their constructs. Nonetheless, based on our understanding of how Dachs works in Hippo versus PCP signaling, there is no indication that their study is inconsistent with the results and model we present here.

*Specific notes: Results:The statements do not make it clear that Ayukawa did see co-IP between Ds and Sple-N. Ayukawa also went a bit further, and showed that binding depended on the very N terminus, and was to a middle region of the Ds ICD.*

We revised the text to note their detection of interaction between Sple-N and Ds-ICD.

*Reduced junction accumulation of Ds in Fat mutants – the authors should also reference Strutt and Strutt 2002, and Ma et al. 2003.*

Thank you for pointing this out, we revised manuscript to include these citations.

*The partial rescue of* fat *mutant abdominal PCP defects by loss of* dachs *has already been published in Mao et al. 2006, admittedly not in the detail presented here.*

This was already cited in the text (subsection ‘Interactions between PCP pathways in the abdomen’, fourth paragraph).

*Similarly, improvement of* ds *mutant wing PCP defects by loss of* dachs *was in Brittle et al. 2012.*

Thank you for pointing this out, we now cite this in the revised manuscript.

*Since* dachs *loss will affect both Sple and the Hippo pathway, can the authors show that the* dachsfat *effect on wing PCP is different than the rescue of* fat *PCP achieved by increasing Hippo pathway activity in* fat *mutants, as claimed in Feng and Irvine, 2007?*

Yes, the effect is different. *dachs* efficiently suppresses both Fat-PCP phenotypes and Fat-Hippo phenotypes. Warts over-expression (as in Feng and Irvine 2007) efficiently suppresses Hippo phenotypes, but these animals still have PCP phenotypes. We now note this in the text and added an example of the wing hair PCP phenotype to the revised manuscript (Figure 4—figure supplement 1).

*"Thus, behind the morphogenetic furrows…" Strictly speaking, the authors have shown that protein polarization depends on Ds, Ft or Sple, but not that these direct it. To do that, they would have to change their levels or, better, repolarize them. They have also not shown that these effects depend on the Ds ICD.*

Agreed, we showed the phenotypes of loss-of-function mutations, and we revised the text accordingly. For the revised manuscript, we attempted to examine the influence of ectopic Ds on Sple localization, however, were unable to do so with available reagents because the frequency of GFP:Sple flip-out was much greater than the frequency of Ds flip-out.

*Methods and figures:The Methods say far too little about how polarity was measured in clones with tagged proteins, only that it was ‘manual’ and used ImageJ.*

We extended Methods in the revised manuscript, which addresses all the questions raised below.

*What was being measured, and how did that translate into the rose plots?*

The intensity of membrane localization of GFP tagged proteins was examined. When membrane accumulation was restricted to one side of a cell, the center of the membrane accumulation was taken as the direction, and this direction was compared to tissue coordinates, as described in the revised Methods. We note that the original text incorrectly referred to the polarity of clones – polarity was actually assessed within individual cells.

*Was there an assessment of the intensity of polarization in a clone, or just a direction for each clone?*

Just direction.

*How was that determined and quantified? Did this assessment only use cell faces at clone boundaries, or internal cell faces?*

The intensity of membrane localization was examined. When membrane accumulation was restricted to one side of a cell, the center of the membrane accumulation was taken as the direction, and this direction was compared to a reference vector as described in the revised Methods.

*Did it adjust for the shapes of the cells and clones, and thus the amount of membrane in each quadrant?*

The direction of polarization was estimated without regard to the sizes or shapes of clones and cells.

*Since proximal and distal cell faces can be at slightly different focal planes due the folding of the pouch, was more than one confocal slice used for the image?*

Yes, multiple confocal slices were combined, and E-cadherin staining was used as a reference to make sure that all sides of a cell were contained within all cells scored.

*And two anatomical questions for the wing disc results: By ‘distal’ and ‘proximal’, do the authors mean the distal tip, or the prospective wing margin?*

As clarified in the revised Methods, within the distal wing, wing margin was used as a reference, because the analysis was limited to regions in which the Fj and Ds gradients are essentially perpendicular to the D-V boundary. Within the proximal wing, proximal wingless expression, which encircles the wing pouch, was used as a reference.

*Second, did they have any objective landmarks for separating proximal from distal portions of the wing pouch? The fate map can be a bit tricky if they are only using the fold, as the tissue close to the fold can change depending on the stage of eversion, especially near the AP boundary. One could always try using the identifiable sensory organ precursors along the third vein. It would be helpful to know where this actually maps to in an adult wing.*

As noted in the Methods, we used the fold at the edge of the wing pouch as a marker, and defined proximal as within 5 cells of the fold. Our analysis was conducted in larval wing discs, before disc eversion. The L3 SOP marker would only provide a point of reference in a tiny portion of the disc, and only at late third instar, and adding it as a marker would require repeating an enormous amount of work, without affecting the conclusions of our study, so we have not done this.

Reviewer #2:

*The manuscript by Ambeganokar and Irvine addressed how two different pathways that regulate planar cell polarity in* Drosophila *are coordinated. Recent work by a number of groups has implicated* pk*/*sple *as being a key in coupling these two pathways and this current manuscript adds to this emerging story. There are several valuable insights that come from this manuscript. A key finding is that Sple can interact physically with both Dachs and Ds. The authors provide a new way to view several issues. They describe a series of genetic experiments that show that Dachs mediates the localization of Sple in wing cells and that in regions where Ds is highly expressed it can independently direct the localization of Sple. They provide evidence for an alternative model for the* ds*/*ft *wing hair phenotype where this is viewed as a consequence of inappropriate* dachs *localization which results in altered accumulation of Sple which can in turn interfere with Pk function. The paper also provides an explanation for several puzzling results from the literature: the ‘rectification’ of the relationship between* fz *and* ds*/*ft *and for the ability of the uniform Ds expression to rescue the hair polarity of* dsfj *mutants. They also describe some interesting new observations that have been missed by others (e.g. that Pk localization is not proximal in all regions of the wing) even if the significance of the observation is unclear. Specific comments: In the subsection ‘Control of Sple polarity in eye discs by Ds-Fat PCP’, Figure 5. It is not clear to me how the authors conclude that the Sple in panel A (left side) is in R4 and in panel D (right side) R3.*

There is a very faint cytoplasmic stain that does not show up in the images but can be detected if the intensity of the green channel is increased. We only scored cells in which GFP:Sple was expressed only in R3 or only in R4.

*Figure 5—figure supplement 2. I did not find this figure to be convincing (particularly B).* We replaced these panels with different examples for the revised manuscript.

*Figure 7 and Figure 9. The location of the spiracle allows the authors to determine the middle of the A compartment in the pleura but it is not clear to me how they determine where the AP boundary is. Are they counting cells?*

Within the tergites, we can determine this by the location and size of hairs. Within the pleura, we can only roughly estimate the location of the A-P boundary, based on comparison to the adjacent tergites (which are present in the original pictures, but mostly cropped out of these panels). We revised the text to point this out.

*In the subsection ‘Competition between polarizing cues’, the authors argue that the previous* pk*/*sple *balance model is not supported by the data and that a competition model is preferable. I agree with the conclusion that we should now think about the* pk*/*sple *interaction as a competition but I see that as providing a mechanism for the balance model.*

We have tried to make this distinction because a requirement for a ‘balance’ implies that you require some of each isoform. While this is true in the fly as a whole, we argue that individual cells only ever require 1 of the isoforms, and that in many places the two isoforms provide distinct, competing, polarity cues, which is conceptually distinct from a requirement for a ‘balance’.

Reviewer #3:

In this manuscript, Ambegaonkar et al. address the function of the Core PCP Pk/Sple locus in regulating the coupling between the Fat and Core PCP systems.

*Earlier work had suggested that Pk and Sple isoforms influence the coupling of the Core PCP and Fat PCP systems. While Sple enforces coupling such that Vang and Ds/Dachs are on the same side of the cell, Pk allows coupling in the opposite orientation or even complete uncoupling of Core and Fat polarity, allowing Core PCP to respond to other orienting cues. Biochemical experiments had suggested that Sple interacts directly with Dachs. These questions had been most thoroughly explored in the wing, but less was known about coupling of Core and Fat PCP in other tissues like the eye and anterior abdomen where Sple function is more important.*

The current manuscript from Ambegaonkar et al. now extends these studies to the eye and abdomen. It also reveals a second interaction between Sple and Dachsous, and shows that Sple is influenced by Dachs and Ds in different places.

*The authors map the localization of Pk and Sple in the wing disc, eye disc and pupal abdomen by generating clones of fluorescently tagged Pk and Sple isoforms that are over-expressed under the control of the actin promoter. They examine how perturbations of Fat system components influence the localization of PkGFP and Sple:GFP, and correlate these with polarity phenotypes. Their data show that PkGFP and Sple:GFP clones have opposite polarity in wing discs and posterior abdomen (where Pk normally dominates), but the same polarity in eye discs posterior to the furrow and the anterior abdomen (where Sple normally dominates). This is an intriguing correlation, but the authors data do not explore or clarify why Pk and Sple localize together in some tissues and oppositely in others. Their data generally confirm the idea that the Fat system orients Sple in the eye disc and wing disc, but the data in the abdomen are very confusing and no clear model emerges for the orientation Sple or Pk here. In general, I think some of the confusion may stem from the clones used to map Sple and Pk polarity, and from the unusual organization of the abdomen – here Sple and Pk-dependent regions are directly opposed to one another. There are several experiments the authors could do to resolve this confusion that should be possible within the time frame of a revision. Pk and Sple polarity:Throughout the, the authors map Pk and Sple polarity using flip out clones of GFP tagged Pk or Sple that are over-expressed under the control of the actin promoter. They use these constructs to deduce the localization of these proteins even in tissues where the particular isoform in question is expressed at much lower levels than the other – for example they use GFP-Sple clones to make conclusions about Sple localization in the wing disc even though endogenous levels are hardly detectable – in any case much lower than Pk. But given the feedback mechanisms that lead to polarized formation of Core PCP domains, and the fact that these proteins are overexpressed, this may not reflect the endogenous localization. What I mean is that the very small amount of Sple that is normally present in the wing disc might ordinarily be incorporated into proximal Core complexes that are oriented by the much larger amounts of Pk and the cues read by Pk. The distal localization of Sple in the wing disc (as visualized by actin promoter-driven over-expression of Sple:GFP) could result from Core PCP reversal caused by Sple over-expression. They conclude that wing cells normally have Pk on one side and Sple on the other, but this may not be true at endogenous levels.*

We think it possible, as the reviewer suggests, that in wild-type wing discs, endogenous Sple is co-localized proximally with Pk and other Fz-PCP proteins. Our view is that PCP is regulated by competing cues, but due to the feedback processes that are intrinsic to PCP systems, normally one cue ‘wins’ and any Pk (and possibly most Sple as well) will co-localize at one side of the cell. What clones (over-)expressing GFP tagged proteins tell us then, is the cues that a particular isoform sees and responds to when it is the predominant isoform in that tissue. We revised the text to comment on this in the Discussion. We further note that antisera that specifically recognize Pk or Sple were reported by David Gubb, and we obtained some of this sera, but he told us that they are unable to recognize endogenously expressed proteins by immunostaining, they can only detect over-expressed proteins, and this was also true in our hands. Moreover, it is usually not possible to define polarization without clonal expression of tagged proteins.

*To rule this out, the authors should stain for other Core PCP components to be sure that these Sple:GFP clones do not disturb the polarity field, and look at adult flies to make sure these clones don't cause polarity defects.*

*If Sple:GFP really localizes proximally in the wing without otherwise disturbing the polarity of Core PCP domains, it would suggest that Sple (unlike Pk) doesn't need other Core PCP proteins to localize to the membrane – not unreasonable if Sple interacts with Ds and Dachs. This would be easy to show by visualizing Sple in the background of* fz *or* vang *mutants – if it is localized by Ds and Dachs, it should still be found on the proximal side of cells in the wing.*

It has already been reported by others that over-expressed Sple disturbs wing polarity, and we have confirmed for the revised manuscript that polarity defects are observed in our hands when GFP:Sple is expressed (Figure 4—figure supplement 1).

We also expect though, that Sple does not necessarily need other Fz-PCP proteins to localize to the membrane, as its distal localization could be directed by Dachs and Ds. For the revised manuscript, we preformed the requested experiment of examining GFP:Sple in a *vang* mutant. The results (Figure 3) show that Sple is still preferentially distal. We also see a relative increase in the cytoplasmic staining for Sple, consistent with published studies describing the influence of Vang (Stbm) on Pk.

*Studies in the abdomen:No clear conclusions emerge from the studies on the abdomen. It is puzzling that* dachs *mutation suppresses mislocalization of Sple caused by loss of* fat *or* ds *in the anterior abdomen. If Sple is normally oriented by* ds *or* dachs*, where does the information now come from?*

We agree that there are aspects of the results in the abdomen that we cannot explain based on our present understanding of the components and their interactions. The most likely explanation for how Sple is localized in *ds dachs* mutants is that Sple responds under these circumstances to the same cues that localize Pk, which localizes anteriorly in *fat* or *ds* mutants as well as *fat dachs* or *ds dachs* double mutants. The question then becomes what localizes Pk (and/or other Fz-PCP components). One answer could be Hedgehog (whose effects have been characterized by Struhl, Lawrence, and Casal), but this could not explain polarity across the whole A compartment, as the Hedgehog gradient flips in the middle. Indeed we think that the flip in polarity in the A compartment in *sple* or *pk sple* mutants could be due to the Fz-PCP pathway becoming directed by the Hh gradient in the absence of Sple. So why doesn’t this happen in *ds dachs*, and what directs polarity in this genotype? We are not sure, but it’s possible as the reviewer suggests below that propagation of polarity between compartments plays a role.

*Furthermore, in the Discussion, the authors say that Fat-Ds signaling modulates Sple but not Pk localization. However their data in the posterior compartment of the abdomen seem to suggest that Pk can also read cues provided by the Fat PCP system (but somehow with the opposite polarity as Sple). There are no identified physical interations between Pk and Fat components and why it should be oriented in this way by Fat PCP in the posterior abdomen but not the anterior abdomen or eye is puzzling. Furthermore, this must not always be true because sometimes Pk is oriented completely orthogonally to Fat PCP (in the pupal wing and in the eye disc before the furrow passes.*

*The abdomen is complex because two regions with different requirements for Pk and Sple lie directly adjacent to one another. Domineering non-autonomous effects of Core PCP likely propagate over some distance through these regions and rescue (or disrupt) Core PCP domain polarity and hair polarity in other regions. This could mask where specific Fat and Core components are actually needed. This could explain why removing Dachs suppresses the polarity defects caused by Fat or Ds mutation in the anterior abdomen – once all signals from Fat PCP are gone, then correct Core polarity, propagating in to the anterior compartment from the posterior compartment, could correctly orient Sple and hair polarity. A similar mechanism might explain why Pk localization in the posterior is disturbed by loss of Fat system components – i.e. Sple-dependent alterations of Core polarity in the anterior are propagating via Core PCP interactions into the posterior.*

*To clarify this, the authors should target RNAi of PCP components to specific subregions of the abdomen to find out where their effects are direct and where they occur as a consequence of domineering non-autonomy.*

We agree that propagation (‘domineering non-autonomy’) of PCP between compartments could potentially influence some of the phenotypes observed (e.g. boundaries of where we see polarity phenotypes often don’t correspond to compartment boundaries). We have attempted over the past two months to perform compartment-specific RNAi as suggested, but have encountered a series of technical difficulties, and found that they are currently not feasible. What we have done instead is to add some text to the Discussion to comment on the establishment of PCP in the abdomen and mention the potential for propagation of polarity between compartments to influence the results.

*More specific comments:1) The authors cite both Hogan et al. and Merkel et al. as saying that there is an early phase of polarity establishment in the wing where Sple is reading a Fat/Ds-dependent signal. But the Merkel et al. paper actually says the opposite – this paper shows that loss of Sple has no consequences for Core PCP orientation in the wing until very late after hair formation when the ridge pattern is generated. The idea that Sple has an early role was based on over-expression experiments.*

Agreed, we have revised the text here.

*2) The cartoon in Figure 1 suggests a proximal distal orientation of the Fat PCP system in the pupal wing, but this is not the case. Four-jointed is expressed around the whole margin and Ds is elevated at the AP boundary as well as the hinge – the polarity pattern of the domains themselves is first radial and then AP oriented in pupal wings. Since the data they show in Figure 1 pertain to the wing disc not the pupal wing, maybe it would be better to show a cartoon of the Fat PCP pattern in the wing disc where it is quite different.*

Agreed, our cartoon here ignored the more complex and dynamic patterns in the pupal wing, so as suggested we replaced this with a cartoon of the wing disc.

*3) In Figure 4, the authors should quantify the hair polarity phenotypes indicating suppression of the fat mutant by* sple *– a single example is not enough.*

We now include this in the Figure 4 legend.

*4) In Figure 6, the authors should indicate the anterior posterior axis on the images. Also, it is important to show wild type polarity rose diagrams with the same breakdown into Af, Am and Ab or A, A* and P as they show for the mutants.*

We have made these suggested changes to the revised manuscript (Figure 6 and supplements).

*5) In the Discussion, I think the authors are making what seems to be a semantic distinction between a ‘balance’ of Pk and Sple (as suggested originally by David Gubb) and ‘competition’ between Pk and Sple supported by their results. The conclusions are really the same – that Pk and Sple antagonize each other and that phenotypic disturbances when one isoform is lost are caused by inappropriate (or unbalanced) activity of the other.*

We agree that the conclusion that they antagonize each other holds, but we think that it is nonetheless an important distinction to make, as a requirement for a ‘balance’ implies that one needs some of each isoform, which is not, as far as we can tell, true for any cell. We revised the text here.

*Additional discussion comments: I would have liked to see more discussion of the similarities and differences between the current manuscript and Ayukawa. Of course it is possible that they do not have any substantial insights into the differences – if so additional comments might not be valuable.*

We don’t feel that we have much to add beyond noting, as we already do, where our observations agree or disagree with Ayukawa et al.

*I also agree the manuscript would be substantially improved by a more detailed discussion of how they measured polarity. A supplemental figure showing an example would be good.*

We have revised the Methods to better explain how polarity was assessed. We could add supplemental figures too, but we think it’s not necessary with the improved description.

*The Strutt et al. paper that describes the Sple:GFP and PkGFP constructs shows that they do disrupt polarity when expressed uniformly in the inappropriate tissues. The Sple:GFP construct causes a* pk*-like wing hair orientation pattern, and the PkGFP construct disturbs omatidial polarity in the eye. I would be surprised if the situation were different in the clones used in this paper, and I think it is essential to check the polarity around these clones. In any case, I think it will be hard to argue that Sple would localize proximally when expressed at endogenous levels in a tissue that is dominated by Pk. The reverse isn't as much of a problem – even though PkGFP is over-expressed in clones, it still generally colocalizes with Sple in the eye and anterior abdomen where Sple dominates. This makes it a very interesting question why PkGFP expression causes polarity defects in the eye as reported by Strutt et al. It would seem to suggest that some function of Pk itself, rather than reversal of Core PCP domains, is responsible for Pk over-expression defects in the eye – what do you guys think?*

We agree that polarity reversals are expected in clones, as previously observed by Strutt et al. and others. In the revised manuscript, we show that PCP phenotypes are induced in the wing by GFP:Sple-expressing clones. In the eye, in order to score PCP we needed ommatidia in which transgenes were expressed only in R3 or only in R4. This was achieved by examining eye discs only 24 hr after clone induction. We estimate that in these eye discs, GFP:Pk was expressed in or behind the furrow, i.e., when and where it co-localizes with Sple. We hypothesized that disturbance of PCP by Pk might require Pk expression in front of the furrow (where it localizes differently than Sple). To test this idea, we compared PCP in eye discs with clones of cells expressing GFP:Pk for one day, versus 3 days. Indeed, we observed normal PCP with 1 day clones, and abnormal PCP with 3 day clones. These results imply that the timing of Pk over-expression is important in determining whether or not a PCP phenotype is induced, and are included in the revised manuscript (Figure 5—figure supplement 2).

*I do also think it is important to target knockdown of different Fat pathway components to specific abdominal regions. At the moment, the results of these experiments are confusing and inconsistent with the results from other tissues – no clear model can be drawn from them. Of course sometimes that's just the way life is, but non-autonomous effects would be a trivial explanation for these discrepancies and I think they should be ruled out so as not to confuse the field needlessly.*

As noted above, we attempted these experiments, but unfortunately were unsuccessful. We revised the text to note the possibility of non-autonomous effects.

*I agree that they need to test the effects of the flip-outs.*

This presumably refers to showing that PCP is disturbed by Flip-out clones expressing Sple and/or Pk? As noted above, we have now added these experiments to the revised manuscript.

*As for the confusing nature of the abdomen, they do provide some possible hypotheses in the Results. In the case of the anterior, Reviewer 3 was asking what supplied the information for normal Sple polarity in the* dsdachs *double. The manuscript opines that it could be whatever localizes Pk, presumable meaning some part of the core system, since Pk is pretty normal. Or I suppose one could invoke Pk-Sple binding. They hypothesized both to explain proximal Sple in* ds dachs *double mutant wings, so it is pretty much the same idea here. The sensitivity of Pk to* ds *or* dachs *removal in the posterior is bit more confusing, as I think they admit. What they suggest is that, in absence of Ds cues, now Pk becomes sensitive to Sple and follows mislocalized Sple. It could be explained a bit better, as the molecular interactions of Dachs and Ds are with Sple; I guess this unbound Sple is now more attractive to Pk? Or we could invoke the weak Pk-Dachs binding of Ayukawa, but it would have to be binding that mediate repulsion or destruction, to get anterior Pk and posterior Dachs. At any rate, they did not look at Pk localization in a* ds sple *or* dachs sple *double, if they are really trying to support this idea. And if we're being complete, they must know what happens to Sple in a* ds dachs *double in the posterior. If Sple gets strongly reversed like it does in* ds dachs *wing, I'm not sure it can account for the Pk mislocalization they note in a* ds dachs *double.*

Sple is predominantly posteriorly localized in abdominal P compartments of *ds dachs* mutants, as it is in wild-type (Figure 6, Figure 8). For revisions we also examined Pk in *dachs sple* (Figure 6—figure supplement 1).

*On the other hand, I would bet in a* ds *or* fat *mutant wing that, if you waited long enough, you would get misorientation of Pk polarization, just like you do of the other core polarity proteins in a 28–32 hr AP wing. What stage of abdomen were they looking at, and how long before hair formation?*

Pk and Sple localization in pleura were examined around 48 hr APF, which is around the time of hair formation. We now note this in the Methods.

*I have to agree that about the need for examining the polarity around the clones. I think that rises to the level of essential. It is hard to imagine that they did not at least look at adult wings for polarity phenotypes.*

This is now included in the revised manuscript. The result (GFP:Sple clones disturb PCP) doesn’t change any of our conclusions.